# Defining the ultrastructure of the hematopoietic stem cell niche by correlative light and electron microscopy

**Sobhika Agarwala[1†], Keun-Young Kim[2†], Sebastien Phan[2], Saeyeon Ju[2], Ye Eun Kong[2], Guillaume A Castillon[2], Eric A Bushong[2], Mark H Ellisman[2,3]*, Owen J Tamplin[1]***

[1]Department of Cell and Regenerative Biology, School of Medicine and Public Health, University of Wisconsin-Madison, Madison, United States; [2]Center for Research in Biological Systems, National Center for Microscopy and Imaging Research, University of California at San Diego, San Diego, United States; [3]Department of Neurosciences, University of California at San Diego School of Medicine, San Diego, United States

**Abstract** The blood system is supported by hematopoietic stem and progenitor cells (HSPCs) found in a specialized microenvironment called the niche. Many different niche cell types support HSPCs, however how they interact and their ultrastructure has been difficult to define. Here, we show that single endogenous HSPCs can be tracked by light microscopy, then identified by serial block-face scanning electron microscopy (SBEM) at multiscale levels. Using the zebrafish larval kidney marrow (KM) niche as a model, we followed single fluorescently labeled HSPCs by light sheet microscopy, then confirmed their exact location in a 3D SBEM dataset. We found a variety of different configurations of HSPCs and surrounding niche cells, suggesting there could be functional heterogeneity in sites of HSPC lodgement. Our approach also allowed us to identify dopamine beta-hydroxylase (dbh) positive ganglion cells as a previously uncharacterized functional cell type in the HSPC niche. By integrating multiple imaging modalities, we could resolve the ultrastructure of single rare cells deep in live tissue and define all contacts between an HSPC and its surrounding niche cell types.

**\*For correspondence:**
mellisman@ucsd.edu (MHE);
tamplin@wisc.edu (OJT)

[†]These authors contributed equally to this work

## Editor's evaluation

The manuscript reports on an extensive body of work, achieving the still highly challenging identification of HSPCs within the ultrastructure of their niche. The study highlights the heterogeneous nature of HSC-niche interactions, which is consistent with heterogeneity identified through genomic and functional studies. The work presented is of high interest to the field.

## Introduction

Hematopoietic stem and progenitor cells (HSPCs) give rise to all blood cell types throughout the life of an organism (*Orkin and Zon, 2008*). HSPCs reside in a complex microenvironment called the niche that is made up of many different kinds of support cells, including various types of mesenchymal stromal cells (MSCs) and endothelial cells (ECs) (*Pinho and Frenette, 2019*). However, our understanding of HSPC interactions with niche cells has been limited to low-resolution light microscopy. Recently developed transgenic reporter lines have allowed identification of well-defined endogenous HSPCs in both mouse and zebrafish model organisms (*Acar et al., 2015*; *Chen et al., 2016*; *Christodoulou et al., 2020*; *Tamplin et al., 2015*). The dynamic behavior of HSPCs in the niche of mouse

and zebrafish can be observed using live imaging (*Bixel et al., 2017*; *Christodoulou et al., 2020*; *Itkin et al., 2016*; *Koechlein et al., 2016*; *Lo Celso et al., 2009*; *Spencer et al., 2014*; *Tamplin et al., 2015*). Yet, it remains challenging to observe HSPC-niche interactions at high resolution. Certain events during hematopoietic ontogeny, such as the colonization of the fetal bone marrow, have so far only been studied in fixed tissues because they are difficult to access (*Coşkun et al., 2014*). Our goal is to better define the ultrastructure of single endogenous HSPCs deep in niche tissue.

We have taken advantage of the transparency and external development of the zebrafish larva to study the earliest migration events of HSPCs into the presumptive adult kidney marrow (KM) niche. We considered using two different correlative light and electron microscopy (CLEM) techniques (*Karreman et al., 2016*) to resolve the ultrastructure of HSPCs. For the first approach, to confirm cell identity we used a label-based approach similar to what was done using an APEX2-Venus-CAAX fusion protein (*Hirabayashi et al., 2018*) that could be visualized both with light microscopy and high contrast in EM imaging. A similar method was applied in zebrafish using an APEX-GBP (GFP Binding Protein) fusion to resolve GFP$^+$ transgene expression on electron micrographs (*Ariotti et al., 2015*). Building on these previous studies, we used genetically encoded APEX2 engineered peroxidase (*Lam et al., 2015*), together with a fluorescent protein, to track single HSPCs as they migrated into and lodged in the larval KM during the earliest colonization stages. For the second approach, we used improved software alignments to merge datasets across different imaging platforms and could correlate single cells without the need for endogenous labels. We found clusters of HSPCs around the glomerulus as previously described, as well as single HSPCs lodged in a perivascular niche. The heterogeneity between sites of HSPC lodgement and their surrounding niche cells supports the model that functionally distinct sites exist within the hematopoietic microenvironment (*Zhang et al., 2021*). These large 3D CLEM datasets also allowed us to identify previously uncharacterized dopamine beta-hydroxylase (dbh) positive ganglion-like cells in the larval kidney niche that were in direct contact with HSPCs.

## Results
## Colonization of the larval kidney niche by circulating HSPCs

To characterize the larval kidney HSPC niche in the zebrafish, we first wanted to determine the location of all HSPCs in the anterior KM. We used HSPC-specific Runx:mCherry$^+$ transgenic zebrafish larvae (*Tamplin et al., 2015*) that were fixed at 5 days post fertilization (dpf) for anti-mCherry immunofluorescence, then optically cleared using benzyl alcohol/benzyl benzoate (BABB). This technique allowed visualization and quantification of all mCherry$^+$ HSPCs in the larvae. We observed bilateral clusters of ~50 HSPCs, for a total of ~100 HSPCs, in the region of the anterior KM at 5 dpf (*Figure 1—figure supplement 1A-D*). During early HSPC colonization of the KM at 4 dpf we observed ~50 total HSPCs, which increased by ~100% from 4 to 5–6 dpf, and then again by ~50% from 5–6 to 7–8 dpf (*Figure 1—figure supplement 1E*). We performed phospho-histone H3 (PH3) antibody labeling to determine if HSPCs were highly proliferative. On average, we only found one to two mitotic HSPCs in the KM at 5 dpf (*Figure 1—figure supplement 1F-J*). Previous studies of hematopoietic cells in the larval KM at 7 dpf also showed low levels of proliferation (*van Rooijen et al., 2009*). These data suggest that the increase in HSPC numbers during the early stages of KM colonization results from arrival via circulation, that is HSPCs that originated in the caudal hematopoietic tissue (CHT) and dorsal aorta (*Murayama et al., 2006*), and not extensive proliferation of resident HSPCs.

To follow the dynamics of HSPC colonization in the KM niche, we performed time-lapse live imaging at 4 and 5 dpf. Although the early zebrafish larva is transparent, imaging live KM using point scanning confocal microscopy is challenging due to the relatively slow acquisition time and depth of the tissue. To rapidly capture HSPC colonization events throughout the entire depth of the larval KM, we performed light sheet fluorescence microscopy (*Huisken and Stainier, 2009*). Using this technique, we could rapidly acquire a Z stack through the entire KM in less than 30 s (>200 slices with 1 μm spacing). Consistent with our observations of fixed embryos, a depth-coded projection of light sheet time-lapse images showed bilateral Runx:mCherry$^+$ HSPC clusters ('green' cells on left and 'blue' cells on right; *Figure 1—figure supplement 2*; *Figure 1—video 1*). Together with the cdh17:GFP reporter line for pronephric tubules (*Zhou et al., 2010*), we localized Runx:mCherry$^+$ HSPCs within the anterior kidney region, mediolateral to the proximal pronephric tubules (*Figure 1—figure supplement 3*). To

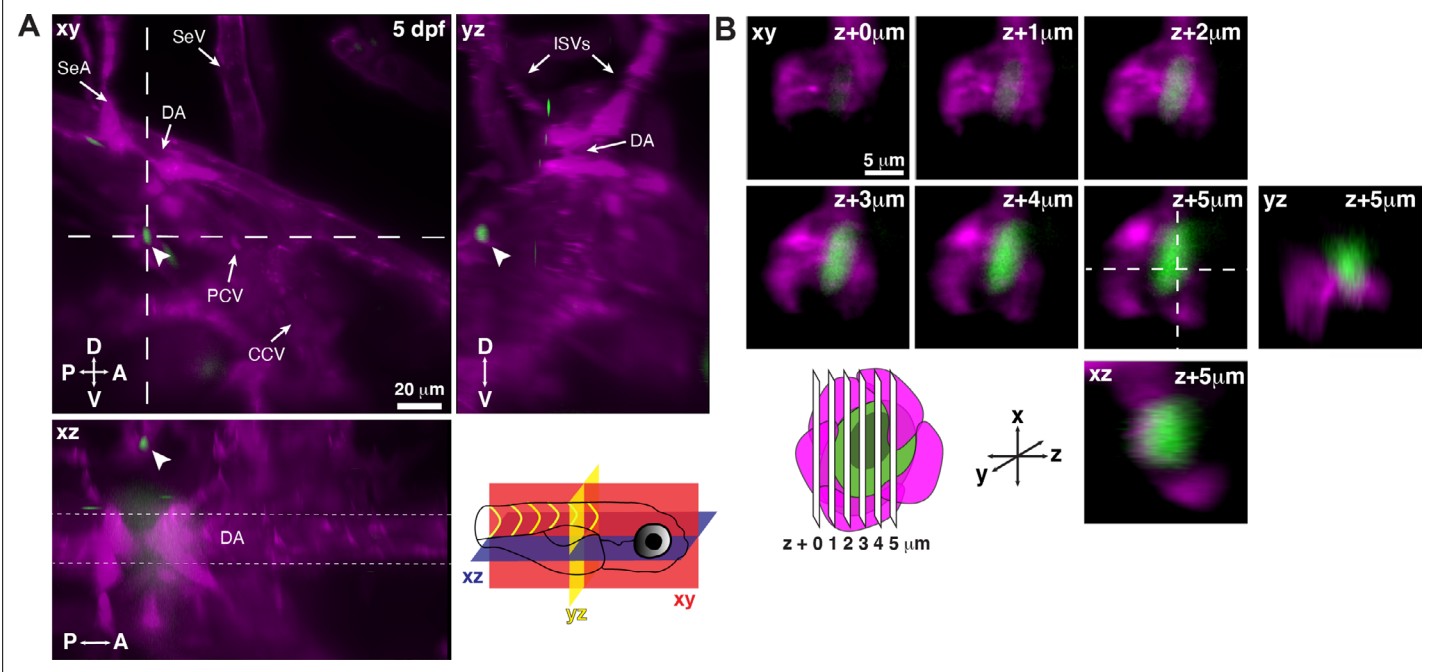

**Figure 1.** Single hematopoietic stem and progenitor cells (HSPCs) lodge in a perivascular region of the larval kidney niche. (**A**) Snapshot of single optical sections (XY, XZ, YZ planes) from light sheet live image of a Runx:GFP;flk:mCherry double transgenic zebrafish larva. A single Runx:GFP+ HSPC (white arrowhead) is lodged in a perivascular region lateral to the dorsal aorta (DA). (**B**) Detail of optical sections (1 µm steps) through the single lodged Runx:GFP+ HSPC in (**A**). mCherry+ endothelial cells contact the HSPC and form a surrounding pocket. The +5 µm section is also shown in XZ and YZ planes. Abbreviations: DA, dorsal aorta; SeA, intersegmental artery; SeV, intersegmental vein; PCV, posterior cardinal vein; CCV, common cardinal vein; ISVs, intersegmental vessels; D, dorsal; V, ventral; A, anterior; P, posterior.

The online version of this article includes the following video and figure supplement(s) for figure 1:

**Figure supplement 1.** Quantification of Runx:mCherry+ HSPCs in the anterior kidney of the zebrafish larva.

**Figure supplement 2.** First frame from light sheet live imaging of Runx:mCherry+ 105 hr post fertilization (hpf) larva.

**Figure supplement 3.** Light sheet live imaging shows hematopoietic stem and progenitor cell (HSPC) clusters are mediolateral to the proximal pronephric tubules (PT).

**Figure supplement 4.** Light sheet live imaging of the anterior kidney reveals hematopoietic stem and progenitor cells (HSPCs) in distinct perivascular regions.

**Figure supplement 5.** Light sheet live imaging of overlap of hematopoietic stem and progenitor cell (HSPC) transgenic reporter lines.

**Figure 1—video 1.** Depth-coded projection of a time-lapse video shows migrating mCherry+ hematopoietic stem and progenitor cells (HSPCs) occupying the kidney marrow (KM) niche.

https://elifesciences.org/articles/64835/figures#fig1video1

**Figure 1—video 2.** Single circulating hematopoietic stem and progenitor cells (HSPCs) interact with the posterior perivascular niche region.

https://elifesciences.org/articles/64835/figures#fig1video2

determine the location of these HSPC clusters relative to the vasculature, we imaged HSPC-specific Runx:mCherry together with the flk:ZsGreen vascular-specific transgenic reporter (*Cross et al., 2003*). HSPC clusters were located between lateral dorsal aortae and cardinal veins (*Figure 1—figure supplement 4*). Light sheet live imaging of the KM niche at 5 dpf allowed us to directly visualize the locations and dynamics of early HSPC niche colonization.

To observe the interaction between single HSPCs as they arrive in the larval KM niche, we imaged the Runx:GFP transgenic line together with flk:mCherry to label vessels (*Figure 1—video 2*). As previously described, Runx:GFP is a more restricted marker of HSPCs and is expressed in the cytoplasm, allowing cellular morphology to be resolved, while Runx:mCherry more broadly labels the progenitor pool and is only localized to the nucleus, making it more appropriate for quantifying single cells (*Tamplin et al., 2015*; *Figure 1—figure supplement 5A*). Upon arrival within the anterior kidney region, rare circulating Runx:GFP+ HSPCs were seen interacting with and lodging in the perivascular

niche (*Figure 1A*). We resolved single lodged HSPCs surrounded by ECs in a pocket-like structure, as we observed previously in the zebrafish CHT and mouse fetal liver (*Tamplin et al., 2015*; *Figure 1B*).

We also examined expression of the cd41:GFP HSPC reporter line in the region of the anterior KM and found substantial overlap with Runx:mCherry⁺ HSPCs (*Figure 1—figure supplement 5A*). The widespread labeling of the 5 dpf HSPC pool by the cd41:GFP reporter, and its cytoplasmic expression that reveals cellular morphology, makes it a valuable tool for further analysis of HSPCs in the larval KM. Furthermore, cd41:GFP⁺ HSPCs were visible in EC pockets, confirming that they had lodged in the KM niche (*Figure 3—figure supplement 1A*). Together, the Runx:mCherry, Runx:GFP, and cd41:GFP reporter lines show similar lodgement in the larval KM niche, and represent an array of valuable tools to interrogate the interaction of HSPCs with various niche cell types.

## HSPCs in the larval KM niche form direct contacts with an MSC and multiple ECs

After characterizing the specific location of HSPCs in the larval KM niche, and their lodgement in EC pocket structures that are also formed in the CHT (*Tamplin et al., 2015*), we sought to identify the specific types of contacts that form between HSPCs and niche cells. We considered that tight junctions may form between HSPCs and ECs and would be marked by tight junction protein 1 (tjp1), the scaffolding protein also known as zonula occludens-1 (ZO-1) that has two orthologues in zebrafish (tjp1a and tjp1b) (*Anderson et al., 1988*; *Stevenson et al., 1986*). We injected Oregon Green dextran into the circulation of Runx:mCherry transgenic larvae to label the vessel lumen, followed by fixation and immunofluorescence with anti-ZO-1 and anti-mCherry antibodies. We observed expression of ZO-1 broadly on ECs, as well as localization between Runx:mCherry HSPCs and surrounding niche cells (*Figure 2A and B*). These data suggest that tight junctions form at the contact points between HSPCs and the niche.

We then went on to characterize additional niche cell types that are present in the larval KM. We performed imaging using the cd41:GFP HSPC transgenic reporter line together with cxcl12:DsRed2 (*Glass et al., 2011*) to label MSCs (*Figure 2C*). We measured the distance between HSPCs and MSCs and observed 57% (n=16/28) of HSPCs were in direct contact with an MSC, 29% were <5 µm (n=8/28), and 14% (n=4/28) were <10 µm away (*Figure 2D*). These data from the larval KM niche are very similar to previous observations from the CHT showing that the majority of HSPCs are in contact with, or close proximity to, an MSC (*Tamplin et al., 2015*). Together, our results demonstrate that HSPC lodgement in the larval KM niche occurs close to or in contact with a single MSC, and that HSPCs are in contact with multiple ECs, similar to what we observed previously in the CHT.

## A CLEM approach to characterize the ultrastructure of HSPCs in the larval KM niche

Next we wanted to explore the ultrastructure of HSPCs and their surrounding cells in the larval KM niche. One approach to analyze the ultrastructure of an entire region of tissue is serial section electron microscopy (EM), a technique that has been used to resolve the projectome of the complete zebrafish larval brain (*Hildebrand et al., 2017*). We previously used CLEM based on anatomical landmarks to match the position of fluorescently labeled HSPCs in the CHT niche between confocal and SBEM datasets (*Tamplin et al., 2015*). However, we found this approach was difficult to apply in the KM because the tissue is much larger and denser than the CHT (Cell Image Library [CIL] accession numbers CIL:54845 and CIL:54850). To confirm precise correlation of single cells between light and EM imaging modalities, we developed two distinct approaches depending on the goals of the experiment. In Workflow #1, we wanted to track lodgement of HSPCs in the larval KM niche using time-lapse live imaging, followed by high-contrast DAB (3,3'-diaminobenzidine) staining with genetically encoded APEX2 to label single endogenous HSPCs in EM sections. In Workflow #2, we wanted to correlate the position of fluorescently labeled cells in existing transgenic reporter lines across confocal and EM datasets. Using these workflows, we were able to observe HSPC lodgement in the KM, then resolve the ultrastructure of those same cells together with the surrounding niche.

The first step in Workflow #1 was to generate a transgenic construct that expressed mCherry for light microscopy, and APEX2 as a genetic tag that allowed electron-dense contrast on target subcellular structures (*Figure 3A*). Our rationale was that mCherry⁺ HSPCs are also APEX2⁺ and will be identifiable by both fluorescence imaging and EM, respectively. We fused APEX2 with H2B and mito

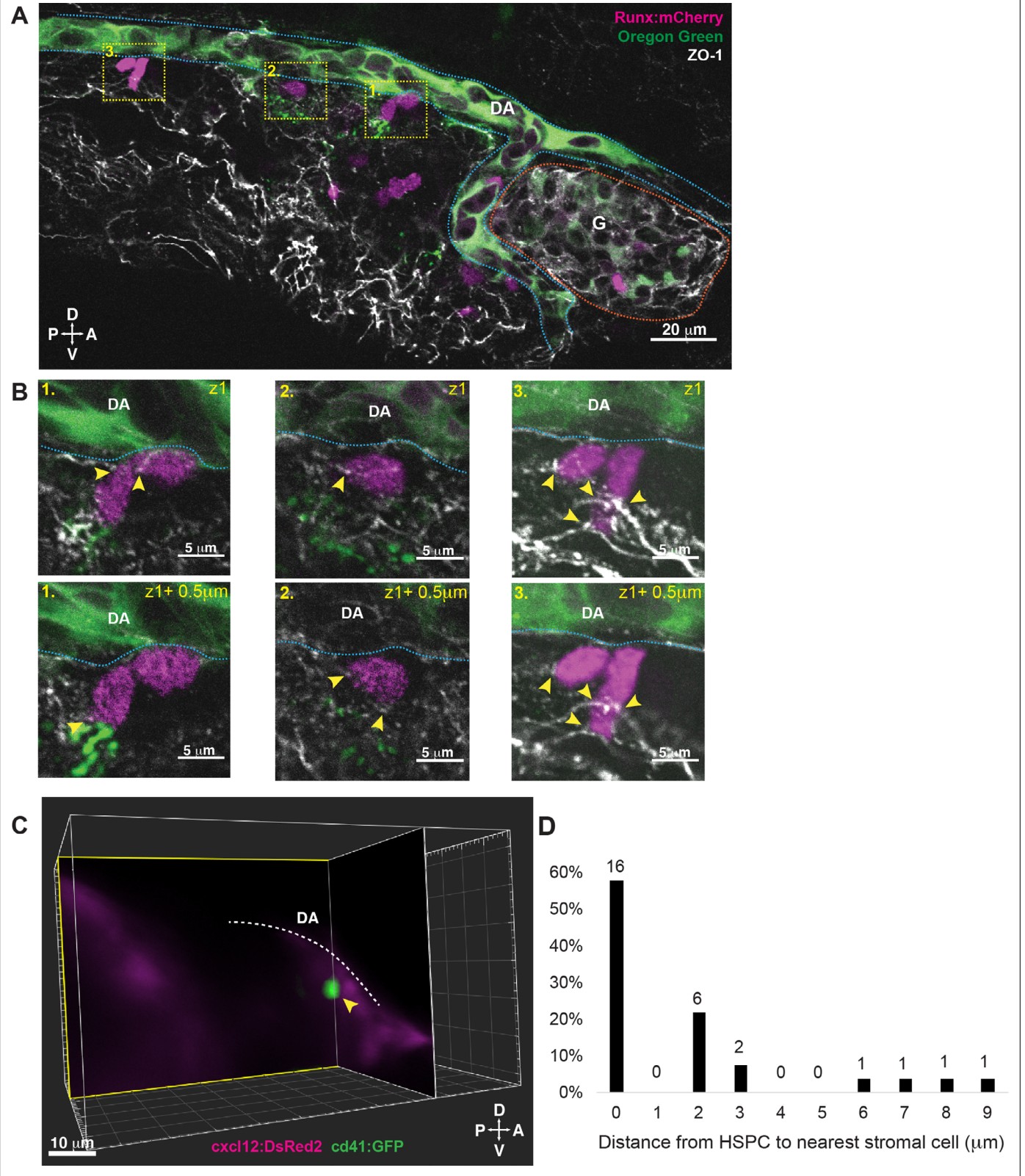

**Figure 2.** Hematopoietic stem and progenitor cells (HSPCs) lodged in the larval kidney niche make direct contacts with endothelial cells (ECs) and mesenchymal stromal cells (MSCs). (**A**) Single optical section from confocal image of larval kidney (fixed) shows Runx:mCherry⁺ HSPCs (magenta) lodged in the perivascular niche. Oregon Green dye labels the vessel lumen. Blue dotted lines surround the dorsal aorta (DA) and red dotted lines surround the glomerulus (G). Tight junction protein is marked by zonula occludens-1 (ZO-1) (white). (**B**) High-resolution optical sections (0.5 µm steps) through

*Figure 2 continued on next page*

*Figure 2 continued*

the boxed regions in (**A**) show ZO-1⁺ contact points between mCherry⁺ HSPCs and the niche (yellow arrowheads). (**C**) Orthogonal slices (XY and YZ planes) from live light sheet 3D volume of larval kidney niche. Single cd41:GFP⁺ HSPCs (green) is in contact or in close proximity (yellow arrowhead) to cxcl12:DsRed2⁺ MSCs (magenta). The white dotted line represents the DA. (**D**) Quantification of distances measured between GFP⁺ HSPC and DsRed2⁺ MSCs shows ~60% of HSPCs are in contact with MSCs, and the remaining are within 9 µm. Numbers above the columns indicate the cell numbers counted in each group (from n=8 embryos). Abbreviations: D, dorsal; V, ventral; A, anterior; P, posterior.

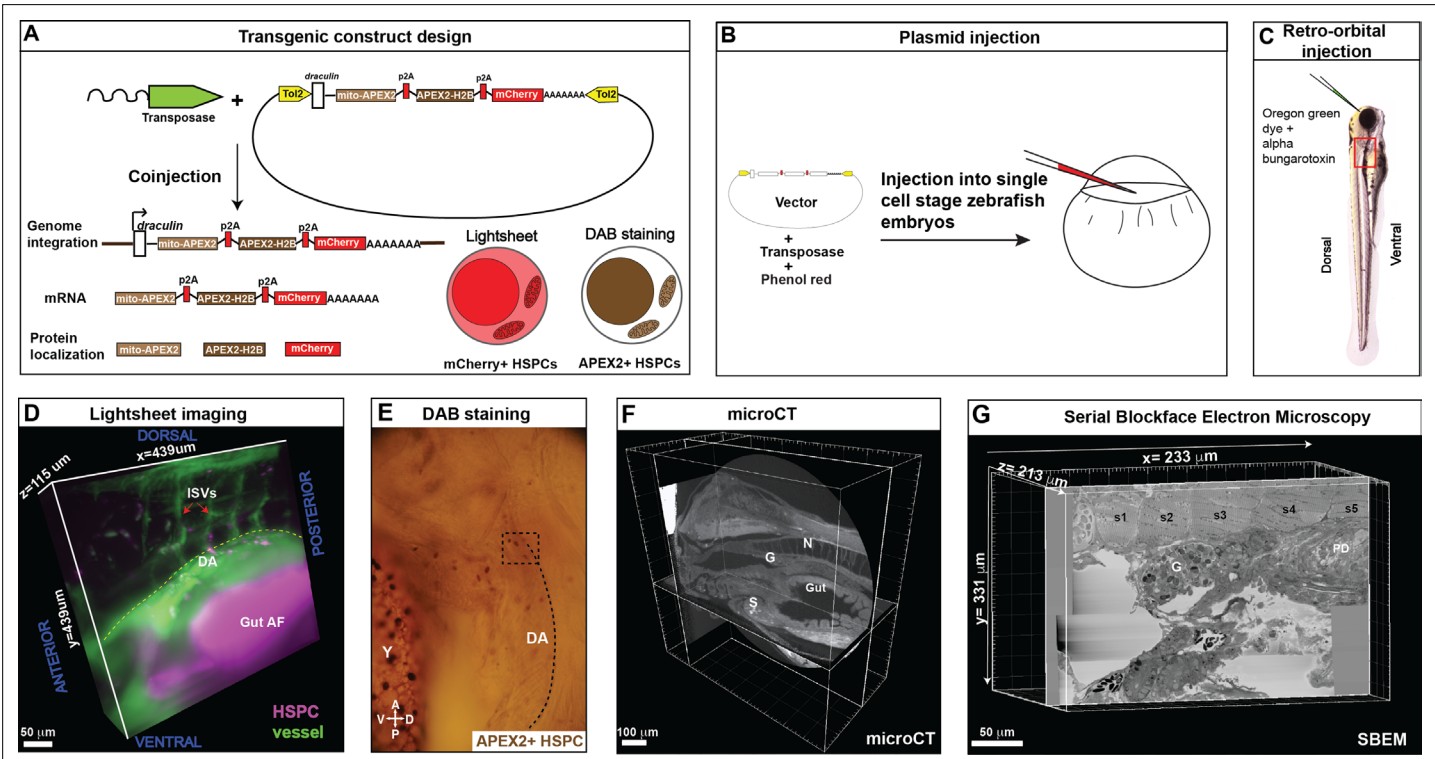

**Figure 3.** Correlative light and electron microscopy (CLEM) Workflow #1 to genetically encode a label in endogenous hematopoietic stem and progenitor cells (HSPCs) for live tracking by light microscopy and high-contrast resolution in serial block-face scanning electron microscopy (SBEM) sections. (**A**) Fusion construct encoding p2A-linked proteins mito-APEX2, APEX2-H2B, and mCherry that localize to the mitochondria, nucleus, and cytoplasm, respectively. The *draculin* promoter was used to transiently drive strong mosaic expression in HSPCs. Random insertion in the genome was by Tol2-mediated transgenesis. (**B**) Tol2 *draculin:mito-APEX2_p2A_APEX2-H2B_p2A_mCherry* (*drl:APEX2-mCherry*) fusion construct was injected together with *tol2* mRNA in one cell wild type zebrafish embryos. (**C**) At 5 days post fertilization (dpf), embryos with circulating mCherry⁺ HSPCs were visually screened and retro-orbitally injected with alpha bungarotoxin to paralyze the embryo, and Oregon Green dye to label the vasculature. (**D**) Dye-injected mCherry⁺ double positive embryos were visually screened and used for light sheet microscopy (example shows a 439 × 439 × 115 µm³ volume of the anterior kidney marrow (KM); ISVs, intersegmental vessels; yellow dotted line, DA, dorsal aorta; gut AF, gut autofluorescence). (**E**) Brightfield example of a single embryo after fixation and DAB (3,3'-diaminobenzidine) staining to label APEX2⁺ HSPCs that are located within the dotted box (dotted line marks DA, dorsal aorta; Y, yolk; D, dorsal; V, ventral; A, anterior; P, posterior). (**F**) After embedding, the sample was oriented and trimmed based on images acquired using micro-computed tomography (microCT) (example shows orthogonal sections in three planes, N; notochord, G; glomerulus, S; swim bladder). (**G**) Single plane from ~3000 sections of SBEM data (example shows a 233 × 331 × 213 µm³ volume; s1-s5, somites 1–5; G, glomerulus; PD, pneumatic duct).

The online version of this article includes the following video and figure supplement(s) for figure 3:

**Figure supplement 1.** Single cd41:GFP⁺ hematopoietic stem and progenitor cell (HSPC) lodged in the perivascular niche.

**Figure supplement 2.** Method for mounting, aligning, and trimming the embedded sample for serial block-face scanning electron microscopy (SBEM).

**Figure 3—video 1.** Transiently labeled hematopoietic stem and progenitor cells (HSPCs) lodge within the perivascular niche.
https://elifesciences.org/articles/64835/figures#fig3video1

**Figure 3—video 2.** Micro-computed tomography (microCT) stack acquired through a fixed zebrafish larva.
https://elifesciences.org/articles/64835/figures#fig3video2

tags for localization to the nucleus and the mitochondrial matrix, respectively. To drive expression of this construct in HSPCs, we cloned these elements under control of the *draculin* (*drl*) promoter, generating the *drl:mito-APEX2_p2A_APEX-H2B_p2A_mCherry* transgene, hereafter referred to as *drl:APEX2-mCherry*. The *drl* promoter is a marker of vascular and hematopoietic lineages (*Herbomel et al., 1999*; *Mosimann et al., 2015*), and we chose it because of its high HSPC expression level compared to other available promoters, such as *Runx1+23* (*Tamplin et al., 2015*; *Nottingham et al., 2007*) or *cd41* (*Ma et al., 2011*). Furthermore, it was previously confirmed that drl:GFP⁺ HSPCs almost completely overlap with Runx:mCherry⁺ HSPCs from embryo to adult (*Henninger et al., 2017*; *Mosimann et al., 2015*).

To track and correlate single cells through multiple imaging modalities, we required sparse labeling of HSPCs. Therefore, we generated transient F0 transgenics with a mosaically labeled HSPC pool. Although a caveat of F0 mosaic transgenics in zebrafish is the inherent variability of labeling between embryos, this allowed us to select embryos that had similar numbers of lodged HSPCs in the KM niche compared to other HSPC reporter lines. We observed only one to two rare HSPCs surrounded by EC pockets in the KM niche of each *cd41:GFP* or *Runx:GFP* transgenic larvae, and selected F0 *drl:APEX2-mCherry* larvae with similar HSPC numbers (*Figures 1 and 3*, *Figure 3—figure supplement 1*). Both *cd41:GFP⁺* and selected F0 *drl:APEX2⁺* larvae had similar numbers of positive HSPCs in the mediolateral clusters of the anterior KM niche (*Figure 3—figure supplement 1B*). We gained further confirmation of HSPC-specific expression in F0 *drl:APEX2-mCherry⁺* larvae by injection of the *drl:APEX2-mCherry* construct into *cd41:GFP* transgenic embryos, and observed ~15% of HSPCs were both mCherry⁺;APEX2⁺ and GFP⁺ (data not shown). These data demonstrate that transient expression of the *drl:APEX2-mCherry* construct in F0 larvae can generate sparse labeling of HSPCs in the KM niche with similar frequency to other stable HSPC-specific reporter lines.

To generate larvae for CLEM Workflow #1, we injected *drl:APEX2-mCherry* construct together with *tol2* transposase into single-cell stage wild type zebrafish embryos (*Figure 3B*). At 5 dpf, larvae with circulating mCherry⁺ HSPCs were injected retro-orbitally with alpha-bungarotoxin and dextran-conjugated Oregon Green dye, to paralyze the larvae and label the vasculature, respectively (*Figure 3C*). Once immobilized and mounted for light sheet live imaging, optical sections were acquired through the entire depth of the KM. A short time-lapse was performed to confirm an HSPC was lodged in the KM and not circulating (~30 min; *Figure 3—video 1*). Single lodged HSPCs could be identified relative to the surrounding tissues (*Figure 3D*), and larvae were immediately fixed after imaging. Larvae were stained with DAB to label APEX2⁺ cells for identification by brightfield microscopy (*Figure 3E*). Larvae were treated with osmium tetroxide and embedded for micro-computed tomography (microCT; *Figure 3F* and *Figure 3—figure supplement 2*; *Figure 3—video 2*). This intermediate microCT step allowed the larvae to be oriented and trimmed to select a discrete region of interest (ROI) for SBEM (*Figure 3G* and *Figure 3—figure supplement 2*). Last, automated SBEM generated over 3000 high-resolution sections of the ROI (e.g., XY = 10 nm/pixel, Z=70 nm/pixel; *Figure 3—figure supplement 2*; *Figure 4—video 1*). Using focal charge compensation (*Deerinck et al., 2018*) which effectively eliminates specimen charging, we obtained a high-resolution SBEM dataset without the need for excessive post-processing alignment. Together, these experimental steps in Workflow #1 allowed us to label an endogenous HSPC that was tracked live, then stained for high-contrast detection in a large SBEM dataset.

To correlate the position of a single labeled HSPC across multiple imaging modalities, we performed 3D software alignment of both light sheet and SBEM datasets. First, we identified anatomical features as landmarks in the SBEM data, such as the somites, glomerulus, and pronephric tubules (*Figure 4—figure supplement 1*). We also observed clustered hematopoietic cells around the glomerulus in the same region as seen by light sheet microscopy (compare *Figure 1—figure supplement 3* and *Figure 4—figure supplement 2*). We merged 3D rendered light sheet and SBEM datasets using image analysis software (Imaris) and aligned matching anatomical features in all three planes (*Figure 4A–C* and *Figure 4—figure supplement 3*). By performing these 3D alignments, we could locate a single APEX2⁺ cell in the SBEM dataset that was <5 μm from the corresponding mCherry⁺ HSPC imaged in the light sheet volume (*Figure 4D*). Furthermore, the APEX2⁺ cell had dark nuclear staining with much higher contrast than any of the surrounding cells, confirming we had identified the same APEX2⁺;mCherry⁺ HSPC across multiple imaging modalities (*Figure 4D*). By correlating 3D light sheet and SBEM data, we could confirm lodgement of a single HSPC in the larval

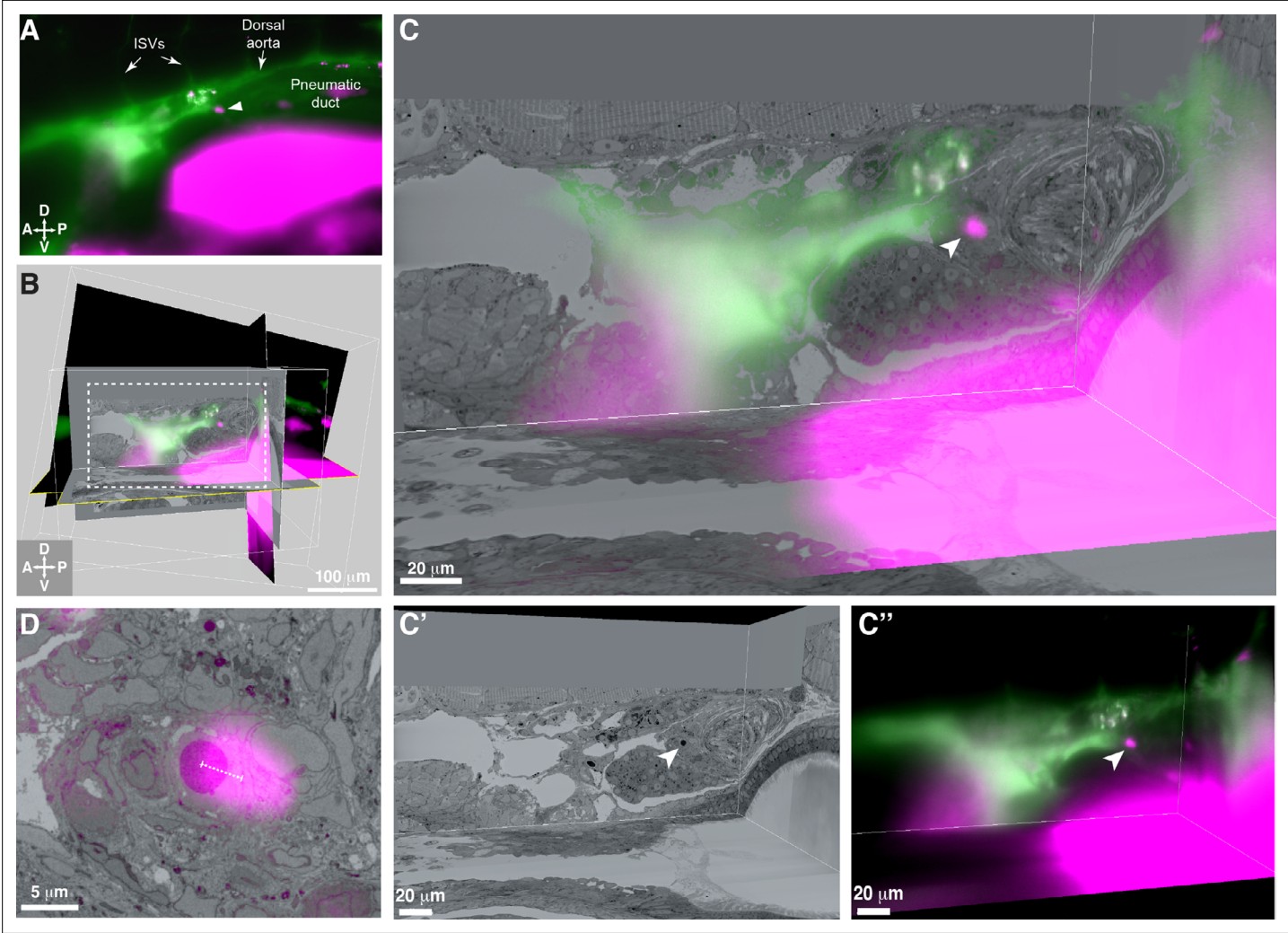

**Figure 4.** 3D alignment of light sheet and serial block-face scanning electron microscopy (SBEM) datasets localizes a single rare hematopoietic stem and progenitor cell (HSPC) across multiple imaging modalities. (**A**) Single Z plane from light sheet imaging of *drl:APEX2-mCherry*+ transgenic larva showing the lodged mCherry+ HSPC (white arrowhead). (**B**) Global alignment of 3D rendered models generated from light sheet and SBEM datasets using Imaris software. (**C**) Orthogonal views of the white boxed region within B shows a 3D view of the alignment between light sheet and SBEM datasets. White arrowhead points to the single lodged HSPC in the aligned light sheet and SBEM datasets. (**C'**) APEX2+ HSPC in SBEM data. (**C"**) mCherry+ HSPC in light sheet data. Green: Injected Oregon Green dextran dye marking vessels. Magenta: Runx:mCherry+ HSPCs and autofluorescence in gut. (**D**) Detail of the alignment shows mCherry+ HSPC and APEX2+ HSPC are <5 μm apart (dotted white line). Abbreviations: ISVs, intersegmental vessels; D, dorsal; V, ventral; A, anterior; P, posterior.

The online version of this article includes the following video and figure supplement(s) for figure 4:

**Figure supplement 1.** Segmentation and 3D surface rendering identifies anatomical features within the serial block-face scanning electron microscopy (SBEM) datasets.

**Figure supplement 2.** Tracing of serial block-face scanning electron microscopy (SBEM) shows location of anterior hematopoietic stem and progenitor cell (HSPC) clusters relative to the glomerulus, pronephric tubules, and cardinal veins.

**Figure supplement 3.** Alignment of 3D rendered models from serial block-face scanning electron microscopy (SBEM) and light sheet live imaging shows anatomical features in all three planes.

**Figure 4—video 1.** All sections (>3000) of a APEX2+ serial block-face scanning electron microscopy (SBEM) dataset.

https://elifesciences.org/articles/64835/figures#fig4video1

KM niche, allowing us to further define the ultrastructure of this rare cell relative to its surrounding cells in the niche.

## 3D modeling of SBEM data reveals all cells in contact with an endogenous HSPC

To reconstruct the spatial relationships of the single APEX2$^+$ HSPC relative to its surrounding niche cells, we performed extensive tracing of cell membranes within the SBEM data using 3D modeling software (IMOD) (*Kremer et al., 1996*). ECs are generally elongated cells with a large nucleus and little cytoplasm, while MSCs can be distinguished by their granular cytoplasm and a nucleus that occupies three-fourths of the cell volume (*Tamplin et al., 2015*). This morphological analysis revealed that the lodged APEX2$^+$;mCherry$^+$ HSPC (*Figure 5A*) was enclosed in a pocket of five ECs, and attached to a single MSC, that all directly contact the surface of the HSPC (*Figure 5B*; *Figure 5—video 1*). This same configuration of cells was seen previously in the CHT (*Tamplin et al., 2015*), demonstrating that this cellular structure is also conserved in the larval KM niche. Within other sections of the SBEM dataset, we observed the clusters of hematopoietic cells posterior to the glomerulus that were also seen with light sheet imaging (compare *Figure 1—figure supplement 3* and *Figure 5—figure supplement 1*).

A significant advantage of SBEM datasets is that they provide a complete 3D picture of the cellular composition of a tissue that is not dependent on prior knowledge of transgenic or immunolabeled markers. Known transgenic markers allowed us to characterize HSPC-EC (*Figure 1*) and HSPC-MSC (*Figure 2C and D*) interactions, but not discover novel HSPC-niche cellular interactions. Careful analysis of our SBEM dataset and the cells in direct contact with the APEX2$^+$;mCherry$^+$ HSPC identified ganglion-like cells in the larval KM (*Figure 5B and C*; *Figure 5—video 1*). The ability to move through all adjacent sections of the 3D SBEM dataset allowed us to follow the length of these ganglion-like cells and discover that it was part of a chain of at least eight morphologically similar cells that extended throughout the larval KM niche.

Further tracing of the neighboring cells in contact with the APEX2$^+$;mCherry$^+$ HSPC revealed two unlabeled cells with the distinctive morphology of putative HSPCs (i.e., scant cytoplasm, large round nucleus, ruffled membrane; HSPC2 and HSPC3; *Figure 5B and C*). These other two putative HSPCs were APEX2 negative, suggesting they were not progeny derived by division from the APEX2$^+$;mCherry$^+$ HSPC, and were more likely independent HSPC clones that had lodged in the same niche. All three HSPCs were in direct contact with the chain of ganglion-like cells that we found extends through the larval KM niche (*Figure 5C*; *Figure 5—video 1*). This 3D SBEM dataset allowed identification of all surrounding cells in contact with the APEX2$^+$;mCherry$^+$ HSPC, and strikingly showed that a single endogenous HSPC can be in direct physical contact with as many as eight other cells.

Finally, given this multicellular HSPC niche structure we observed, we reasoned that an unlabeled HSPC in an independent dataset should be identifiable based on location and morphology alone. We generated a second SBEM dataset that also had a single APEX2$^+$;mCherry$^+$ HSPC, however it was found in a vessel lumen in the larval KM niche and attached to the vessel wall (*Figure 5—figure supplement 2*). We searched the perivascular regions of the larval KM niche within this second SBEM dataset and found two unlabeled putative HSPCs. Not only did both putative HSPCs share a distinct morphology, but following 3D modeling of all surrounding niche cells in contact with the HSPCs, we found each one was also in its own pocket of five ECs, and attached to a single MSC, exactly as we had observed previously in the CHT (*Tamplin et al., 2015*), and with the APEX2$^+$;mCherry$^+$ labeled HSPC (compare *Figure 5C* and *Figure 5—figure supplement 3*). Furthermore, one of the unlabeled putative HSPCs was also in contact with a chain of ganglion-like cells (*Figure 5—figure supplement 3*). In summary, using just the anatomical location and distinctive morphology of an HSPC, we could reconstruct the 3D niche configuration of two more putative HSPCs in the larval KM niche.

## A second CLEM approach to characterize fluorescently labeled cells in the larval KM niche

While the APEX2$^+$;mCherry$^+$ labeling and CLEM approach developed in Workflow #1 was effective for characterizing individual cells lodged in the larval KM niche, we also wanted an approach to characterize all cells in the region that were labeled by an existing fluorescent transgenic reporter line. Toward that goal, we developed Workflow #2 for CLEM analysis of GFP$^+$ cells in thick sections of the larval kidney niche (*Figure 6*). Briefly, GFP$^+$ transgenic larvae were fixed, sectioned using a vibratome,

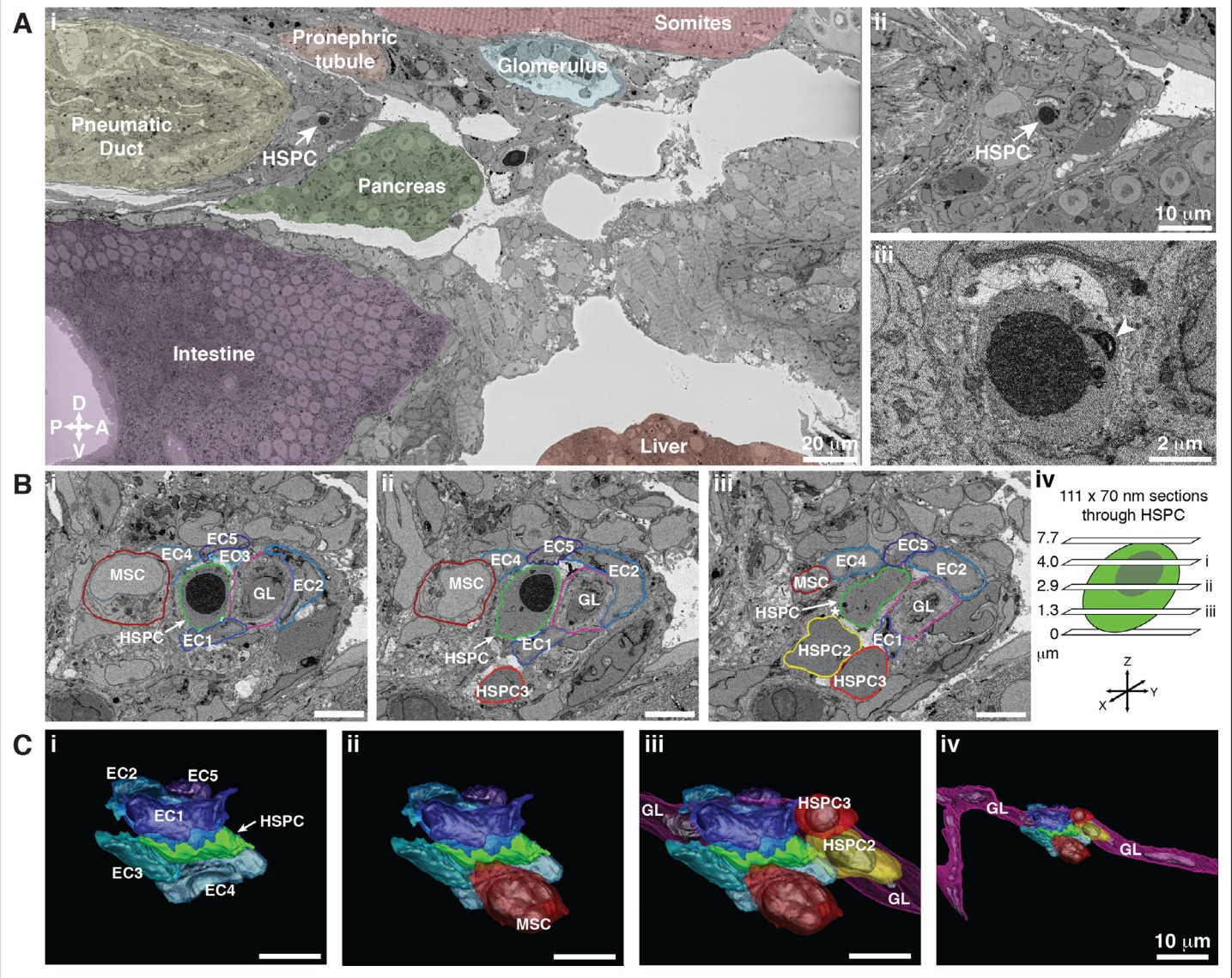

**Figure 5.** Hematopoietic stem and progenitor cells (HSPCs) lodge in a multicellular niche in the perivascular kidney marrow (KM). The ultrastructure of a single APEX2+ HSPC (white arrow) and its surrounding niche cells are modeled using 3D SBEM (00:15 from *Figure 4—video 1*). (**A**) The APEX2+ HSPC is lodged in the perivascular KM niche. (**Ai**) Surrounding tissues are labeled; the HSPC is anterior to the pneumatic duct, dorsal to intestine and pancreas, and ventral to the somites and pronephric tubule. (**Aii**) Higher magnification shows the APEX2+ HSPC is only two-cell diameters from the vessel lumen (white area). (**Aiii**) Full resolution detail of the APEX2+ HSPC showing high-contrast labeling of the nucleus (APEX2-H2B), mitochondria (mito-APEX2; white arrowhead), and extracellular space dorsal to the cell. (**B**) (**i–iii**) SBEM sections at different levels through the APEX2+ HSPC (white arrows) as shown in the schematic (**iv**). The HSPC is simultaneously in contact with multiple niche cells: five endothelial cells (EC1–5), 1 mesenchymal stromal cell (MSC), and a ganglion-like (GL) cell. Two unlabeled APEX2 negative putative HSPCs were lodged in the same niche (HSPC2 and HSPC3). HSPC2 is attached to HSPC3, and the APEX2+ HSPC (**Biii**; asterisk). (**C**) 3D rendered models of the APEX2+ HSPC (solid green) in contact with niche cells. 3D contours are in the same colors as outlines in (**B**). The APEX2+ HSPC is directly contacted by: (**i**) five ECs; (**ii**) one MSC; (**iii**) one HSPC, and a chain of GL-like cells. (**iv**) The GL-like cell is part of a long continuous chain of similar cells that extends through the niche. Scale bars: 5 μm unless otherwise labeled. Abbreviations: D, dorsal; V, ventral; A, anterior; P, posterior.

The online version of this article includes the following video and figure supplement(s) for figure 5:

**Figure supplement 1.** Hematopoietic clusters are visible in the serial block-face scanning electron microscopy (SBEM) dataset.

**Figure supplement 2.** 3D alignment of a second serial block-face scanning electron microscopy (SBEM) dataset showing localization of a single hematopoietic stem and progenitor cell (HSPC) across multiple imaging modalities.

**Figure supplement 3.** Tracing of unlabeled putative hematopoietic stem and progenitor cells (HSPCs) in a second serial block-face scanning electron microscopy (SBEM) dataset shows their relationship with niche cells.

*Figure 5 continued on next page*

*Figure 5 continued*

**Figure 5—video 1.** Hematopoietic stem and progenitor cells (HSPCs) lodge in a multicellular niche structure in the perivascular kidney marrow (KM).
https://elifesciences.org/articles/64835/figures#fig5video1

incubated with DRAQ5 fluorescent nuclear dye, and imaged by confocal microscopy. After imaging, DAB staining solution was added to the sections, followed by illumination to photooxidize DRAQ5. This step locally generates reactive oxygen species that trigger DAB polymerization and darkening of the nuclei (*Ou et al., 2017*). As in Workflow #1, samples were embedded, microCT was used to orient and trim the block, followed by SBEM. Confocal microscopy, microCT, and SBEM datasets are aligned in 3D using Imaris software.

Having established Workflow #2 to perform CLEM using existing GFP+ transgenic lines, we generated a dataset for the KM niche of a cd41:GFP+ larva. We chose the cd41:GFP+ transgenic line because of its consistent labeling of HSPCs in mediolateral clusters (*Figure 1—figure supplement 5*) and lodgement of HSPCs in EC pockets (*Figure 3—figure supplement 1*). This allowed us to characterize the different configurations of many individual HSPCs and their surrounding niche cells, as well as compare these data from an established HSPC-specific reporter line with our other datasets. After brief fixation we could still see a small cluster of cd41:GFP+ HSPCs in the region of the anterior KM niche (*Figure 6A*). Confocal imaging was performed on a thick 100 µm vibratome section after DRAQ5 staining (*Figure 6B*). After embedding, microCT data was used to orient the sample for SBEM. Confocal and microCT datasets were aligned based on general nuclear staining of the tissue (*Figure 6C*). After SBEM, Imaris software was used to align confocal and SBEM datasets in 3D (*Figure 6D and E*). Multiple cd41:GFP+ HSPCs in the larval KM niche were precisely aligned across these different imaging modalities at the single-cell level (*Figure 6F*).

To further confirm the 3D alignment of all cd41:GFP+ HSPCs across imaging platforms, we compared the position of predicted cells by software analysis of the confocal Z stack (using Imaris 'Spots' function), with segmentation of putative HSPCs based on morphology in the SBEM dataset (using Imaris manual 'Surfaces' tracing function). The software predicted 10 cd41:GFP+ HSPCs in the confocal Z stack, and all of those cells overlapped with traced models of the nearest putative HSPC in the SBEM dataset (*Figure 6—figure supplement 1*). Interestingly, extensive tracing of putative HSPCs in the SBEM dataset revealed an additional seven cells that were not well defined by their GFP signal in the confocal dataset (*Figure 6—figure supplement 2*). These putative HSPCs were all deeper in the sample, so were either GFP- or not detected because of low GFP signal.

All 17 putative HSPCs in the larval KM niche (GFP+, GFP-, or GFPlow) were classified based on their location (i.e., within a hematopoietic cluster, vessel lumen, or perivascular niche), and by their cellular contacts (i.e., EC, MSC, HSPC, etc.). These data were summarized together with all putative HSPCs found by SBEM in this study (n=22; *Table 1*). Regardless of their location, all putative HSPCs were in contact with between 2 and 6 ECs. Strikingly, 50% of putative HSPCs were in contact with a single MSC (n=11/22; *Table 1*), similar to what we found by light microscopy in the larval KM niche (60% cd41:GFP+ HSPCs in contact with cxcl12:DsRed2+ MSC; *Figure 2*), and previously in the CHT (60% Runx:GFP+ HSPCs in contact with cxcl12:DsRed2+ MSC; *Tamplin et al., 2015*). Our second CLEM workflow enabled analysis of all cd41:GFP+ HSPCs within an SBEM dataset, and allowed identification of common features shared by many putative HSPCs. Considering all our SBEM data together (*Table 1*), we have also identified considerable heterogeneity between the different structural configurations of a single HSPC and its surrounding niche cells. This suggests that there could be functional heterogeneity as well between different sites of hematopoiesis in the microenvironment. Our current SBEM approaches are not high throughput, so acquisition of more replicates in future studies will better inform the extent of structural heterogeneity between sites of HSPC lodgement in the niche.

## Ganglion-like cells discovered by SBEM are dbh positive niche cells

Our SBEM datasets allowed us to identify all cells in contact with putative HSPCs and discover uncharacterized ganglion-like cells in the larval KM niche. To determine the identity of the ganglion-like niche cells that were in contact with HSPCs (*Figure 5* and *Figure 5—figure supplement 3*), we searched for candidate neuronal cells present in the kidney region at 5 dpf, and identified dbh positive ganglion cells to be the most likely candidate (*An et al., 2002*; *Guo et al., 1999*; *Stewart et al., 2006*; *Stewart et al., 2004*; *Zhu et al., 2012*). Previous studies have shown that dopamine signaling regulates HSPC

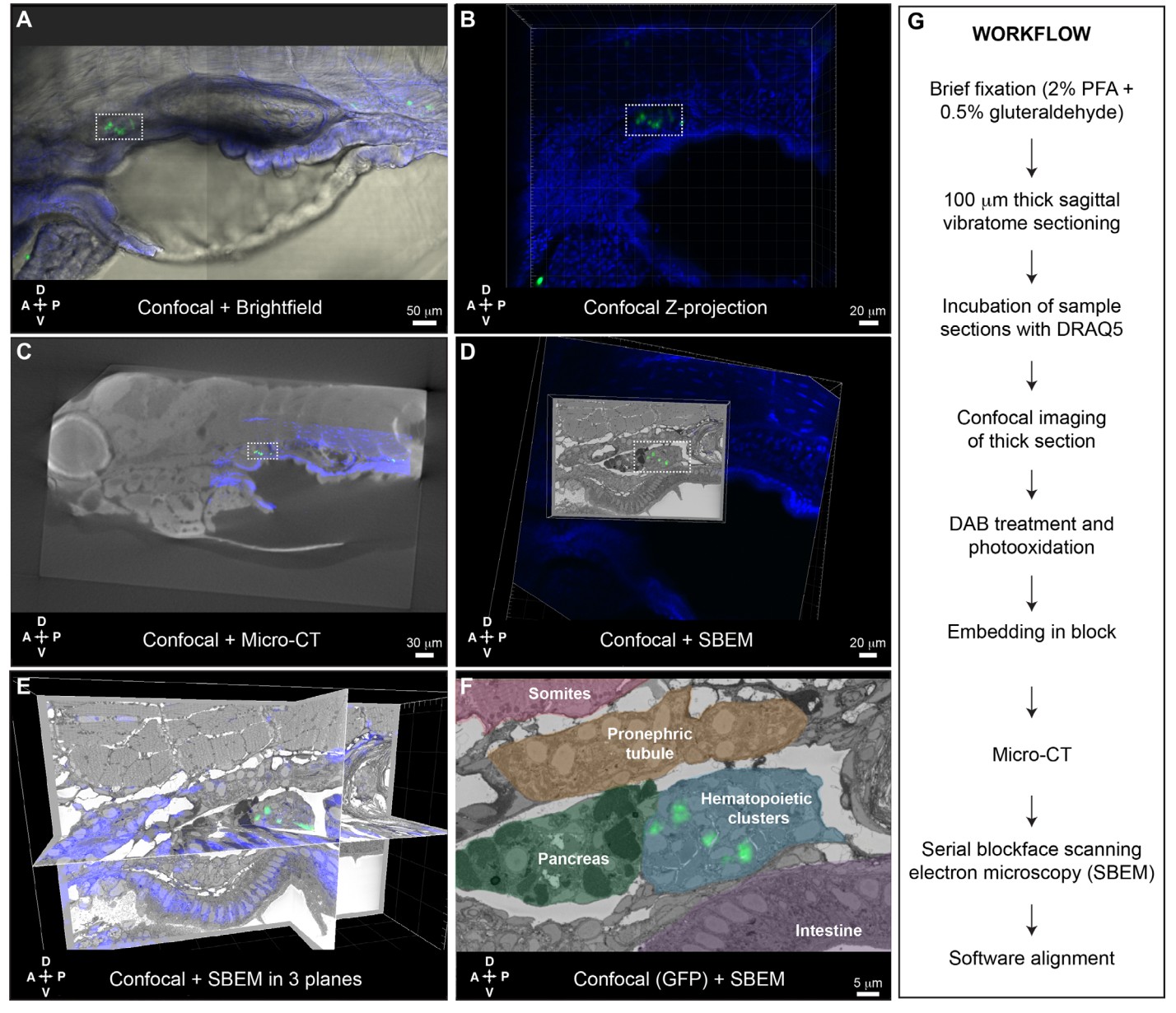

**Figure 6.** Correlative light and electron microscopy (CLEM) Workflow #2 to align all cd41:GFP+ hematopoietic stem and progenitor cells (HSPCs) in the larval kidney marrow (KM) niche. Five days post fertilization (dpf) cd41:GFP+ HSPCs (green) and DRAQ5 nuclear dye (blue). (**A-D**) The same region of the KM niche is marked by a white dotted rectangle. (**A**) Confocal and brightfield image of thick vibratome section. (**B**) Confocal Z projection of thick vibratome section. (**C**) Aligned overlay of micro-computed tomography (microCT) and confocal data. (**D**) Aligned overlay of serial block-face scanning electron microscopy (SBEM) and confocal data (XY plane only). (**E**) Aligned overlay of SBEM and confocal data (XY, XZ, YZ planes). (**F**) Detail of single SBEM section with aligned overlay of cd41:GFP+ HSPCs. Anatomical features are labeled and color-coded. (**G**) Summary of processing steps used in Workflow #2. Abbreviations: D, dorsal; V, ventral; A, anterior; P, posterior.

The online version of this article includes the following figure supplement(s) for figure 6:

**Figure supplement 1.** Positional overlap of cd41:GFP+ hematopoietic stem and progenitor cells (HSPCs) in confocal data and segmentation of putative HSPCs in serial block-face scanning electron microscopy (SBEM) data.

**Figure supplement 2.** Segmentation of cd41:GFP+ and cd41:GFP- putative hematopoietic stem and progenitor cells (HSPCs) in serial block-face scanning electron microscopy (SBEM) data.

**Table 1.** Summary of all putative hematopoietic stem and progenitor cells (HSPCs) in serial block-face scanning electron microscopy (SBEM) datasets, their locations, and niche cell contacts.

'Cell Image Library (CIL)' refers to the public database accession numbers. 'SBEM dataset' refers to one of the four datasets used in the study: APEX2 #1, APEX2 #2, cd41:GFP, and dbh:GFP. 'Figure' refers to the figure where the putative HSPC is shown in the manuscript. 'Putative HSPC' refers to the number of the HSPC as it is annotated within its respective figure. 'Label' refers to the endogenous genetic label for HSPCs in each sample. Label is not applicable (N/A) in the dbh:GFP sample because GFP marks the niche cells and not the HSPCs. 'Color code' refers to the outline, 3D model, or color overlay used to distinguish between different putative HSPCs in the SBEM dataset. 'Location' describes where the putative HSPC is located within the niche: perivascular lodgement, in a vessel lumen, or in the hematopoietic clusters adjacent to the glomerulus. Cell contact columns indicate how many individual niche cells (based on morphology, except in the case of dbh:GFP) are in direct contact with a single putative HSPC.

| Cell Image Library (CIL) | SBEM dataset | Figure | Putative HSPC | Label | Color code | Location | EC contact | MSC contact | RBC contact | HSPC contact | GL contact |
|---|---|---|---|---|---|---|---|---|---|---|---|
| CIL:54847 | APEX2 #1 | 5 | 1 | APEX2+ | Green | Perivascular | 5 | 1 | 0 | 2 | 1 |
| CIL:54846 | APEX2 #2 | 5-S2 | 1 | APEX2+ | None | Lumen | 2 | 0 | 0 | 0 | 0 |
| CIL:54846 | APEX2 #2 | 5-S3 | 1 | APEX2- | Green | Perivascular | 5 | 1 | 0 | 0 | 1 |
| CIL:54846 | APEX2 #2 | 5-S3 | 2 | APEX2- | Green | Perivascular | 5 | 1 | 0 | 0 | 0 |
| CIL:54849 | cd41:GFP | 6 | 1 | GFP+ | Red | Cluster | 5 | 0 | 0 | 0 | 0 |
| CIL:54849 | cd41:GFP | 6 | 2 | GFP+ | Orange | Lumen | 3 | 0 | 3 | 0 | 0 |
| CIL:54849 | cd41:GFP | 6 | 3 | GFP+ | Yellow | Cluster | 5 | 1 | 2 | 2 | 0 |
| CIL:54849 | cd41:GFP | 6 | 4 | GFP+ | Lime | Cluster | 3 | 0 | 1 | 0 | 0 |
| CIL:54849 | cd41:GFP | 6 | 5 | GFP+ | Bright green | Cluster | 5 | 1 | 0 | 0 | 0 |
| CIL:54849 | cd41:GFP | 6 | 6 | GFP+ | Teal | Cluster | 3 | 0 | 3 | 0 | 0 |
| CIL:54849 | cd41:GFP | 6 | 7 | GFP+ | Light blue | Cluster | 5 | 0 | 1 | 0 | 0 |
| CIL:54849 | cd41:GFP | 6 | 8 | GFP+ | Dark blue | Lumen | 3 | 0 | 1 | 0 | 0 |
| CIL:54849 | cd41:GFP | 6 | 9 | GFP+ | Magenta | Lumen | 3 | 0 | 2 | 0 | 0 |
| CIL:54849 | cd41:GFP | 6 | 10 | GFP+ | Dark red | Cluster | 5 | 1 | 1 | 1 | 0 |
| CIL:54849 | cd41:GFP | 6 | 11 | GFP low | White | Cluster | 2 | 1 | 1 | 3 | 0 |
| CIL:54849 | cd41:GFP | 6 | 12 | GFP low | White | Cluster | 3 | 1 | 1 | 0 | 0 |

*Table 1 continued on next page*

*Table 1 continued*

| Cell Image Library (CIL) | SBEM dataset | Figure | Putative HSPC | Label | Color code | Location | EC contact | MSC contact | RBC contact | HSPC contact | GL contact |
|---|---|---|---|---|---|---|---|---|---|---|---|
| CIL:54849 | cd41:GFP | 6 | 13 | GFP low | White | Cluster | 6 | 0 | 1 | 0 | 0 |
| CIL:54849 | cd41:GFP | 6 | 14 | GFP low | White | Cluster | 4 | 1 | 1 | 0 | 0 |
| CIL:54849 | cd41:GFP | 6 | 15 | GFP- | White | Cluster | 5 | 0 | 1 | 1 | 0 |
| CIL:54849 | cd41:GFP | 6 | 16 | GFP- | White | Cluster | 5 | 0 | 2 | 0 | 0 |
| CIL:54849 | cd41:GFP | 6 | 17 | GFP low | White | Cluster | 5 | 1 | 2 | 0 | 0 |
| CIL:54848 | dbh:GFP | 7 | 1 | None | Magenta | Perivascular | 5 | 1 | 0 | 1 | 1 |

EC = endothelial cell. MSC = mesenchymal stromal cell. RBC = red blood cell. HSPC = hematopoietic stem and progenitor cell. GL = ganglion-like cell.

function in the mammalian bone marrow (*Afan et al., 1997*; *Katayama et al., 2006*; *Liu et al., 2021*; *Méndez-Ferrer et al., 2008*). We crossed the dbh:GFP (*Zhu et al., 2012*) and Runx:mCherry transgenic lines, and observed dbh:GFP⁺ cell projections in contact with or in close proximity to mCherry⁺ HSPCs (*Figure 7A and B*; *Figure 7—figure supplement 1A*).

To understand the functional significance of dopamine signaling during KM niche colonization, we treated Runx:mCherry⁺ larvae with 6-hydroxydopamine (6-OHDA), a neurotoxin that induces lesions in dopaminergic neurons (*Jackson-Lewis et al., 2012*; *Matsui and Sugie, 2017*; *Vijayanathan et al., 2017*). We confirmed the efficacy of the drug treatment by reduced locomotor activity (*Feng et al., 2014*) of the larvae (data not shown). A previous study indicated that disrupting dopamine signaling during early stages of development (five somite stage to 30 hours post fertilization (hpf)) resulted in reduced HSPC numbers within the CHT at 48 hpf (*Kwan et al., 2016*). Since our aim was to examine the effect of disrupting dopamine signaling on HSPCs during KM niche colonization, we treated larvae with 6-OHDA from 4 to 5 dpf during KM niche colonization. We observed a significant reduction in the number of Runx:mCherry⁺ HSPCs within the niche after treatment with this dopaminergic cell neurotoxin (*Figure 7C*; *Figure 7—figure supplement 1C, D*).

Last, we validated the presence of dbh⁺ cells within the KM niche by confocal imaging of dbh:GFP⁺ transgenic larva, followed by SBEM, and then correlation of the two datasets using our CLEM approach to analyze any GFP⁺ transgenic line (*Figure 6G*; Workflow #2). dbh:GFP⁺ cells aligned with ganglion-like cells in the KM niche. We identified one ganglion-like cell in contact with a lodged cell that morphologically resembled an HSPC, with the caveat that this cell lacks an independent marker of HSPC identity (*Figure 7D*; *Figure 7—figure supplement 1B*; *Figure 7—videos 1; 2*). We have confirmed that dbh⁺ ganglion-like cells are present in the larval KM niche and are in direct contact with putative HSPCs. Furthermore, we used a small molecule approach to functionally test the role of dopamine signaling in HSPC colonization of the larval KM niche.

## Discussion

We developed two novel CLEM workflows to study the ultrastructure of HSPCs in the larval KM niche. In the first (Workflow #1), we used a genetically encoded fluorescent reporter (i.e., mCherry) together with APEX2 that produces a high-contrast label in SBEM. An advantage of this approach was that we could label specific organelles with a dark stain on EM sections (e.g., mitochondria and nuclei). In a second approach (Workflow #2), we could use any GFP⁺ transgenic line to match the position of a single labeled cell across confocal and EM platforms. In all of the SBEM datasets, we could also identify putative unlabeled HSPCs based on their distinct size and morphology (i.e., 6–7 µm diameter, round, ruffled membrane, large nucleus, scant cytoplasm). In all datasets, lodged putative HSPCs made similar contacts with niche cells: two to six ECs, zero or one MSC, and sometimes a red blood cell, another HSPC, and/or a ganglion-like cell. Strikingly, all of these niche cells could simultaneously contact a single HSPC. This highlights the complexity of signals that an HSPC could receive, as well as the challenges ahead to resolve the functional significance of each contact on stem cell regulation.

In the zebrafish larva, dbh⁺ sympathetic ganglion cells are found in the larval KM region at 5 dpf (*An et al., 2002*; *Stewart et al., 2006*; *Stewart et al., 2004*; *Zhu et al., 2012*). We considered that these cells could be the ganglion-like cells that we found in direct contact with HSPCs. We used additional rounds of CLEM to confirm that dbh:GFP⁺ cells were adjacent to HSPCs in the perivascular KM niche. Chemical inhibition of dbh during KM colonization significantly reduced the number of lodged HSPCs. Our findings are consistent with the established role for neuronal regulation of HSPCs in zebrafish and mammals (*Agarwala and Tamplin, 2018*). Intriguingly, these dbh⁺ cells originate from the same neural crest lineage as a subtype of fetal bone marrow MSCs (*An et al., 2002*; *Isern et al., 2014*). Together, our results have demonstrated that CLEM is a viable approach to identify single rare stem cells deep in live tissue, and that 3D models of a stem cell in its niche built from SBEM data provides a complete picture of stem cell-niche interactions.

The identification and characterization of endogenous stem cells in complex tissues remains extremely challenging. Taking advantage of the transparent zebrafish larva and light sheet imaging, we found it possible to track a single putative HSPC deep in live tissue. We directly observed the larval KM niche during the earliest stages of HSPC colonization (4–5 dpf). This is comparable to HSPC seeding of the fetal bone marrow in mammals. Multiple HSPC-specific reporter lines (e.g., cd41:GFP, Runx:GFP, Runx:mCherry), as well as transient labeling using the *drl* promoter, all marked putative

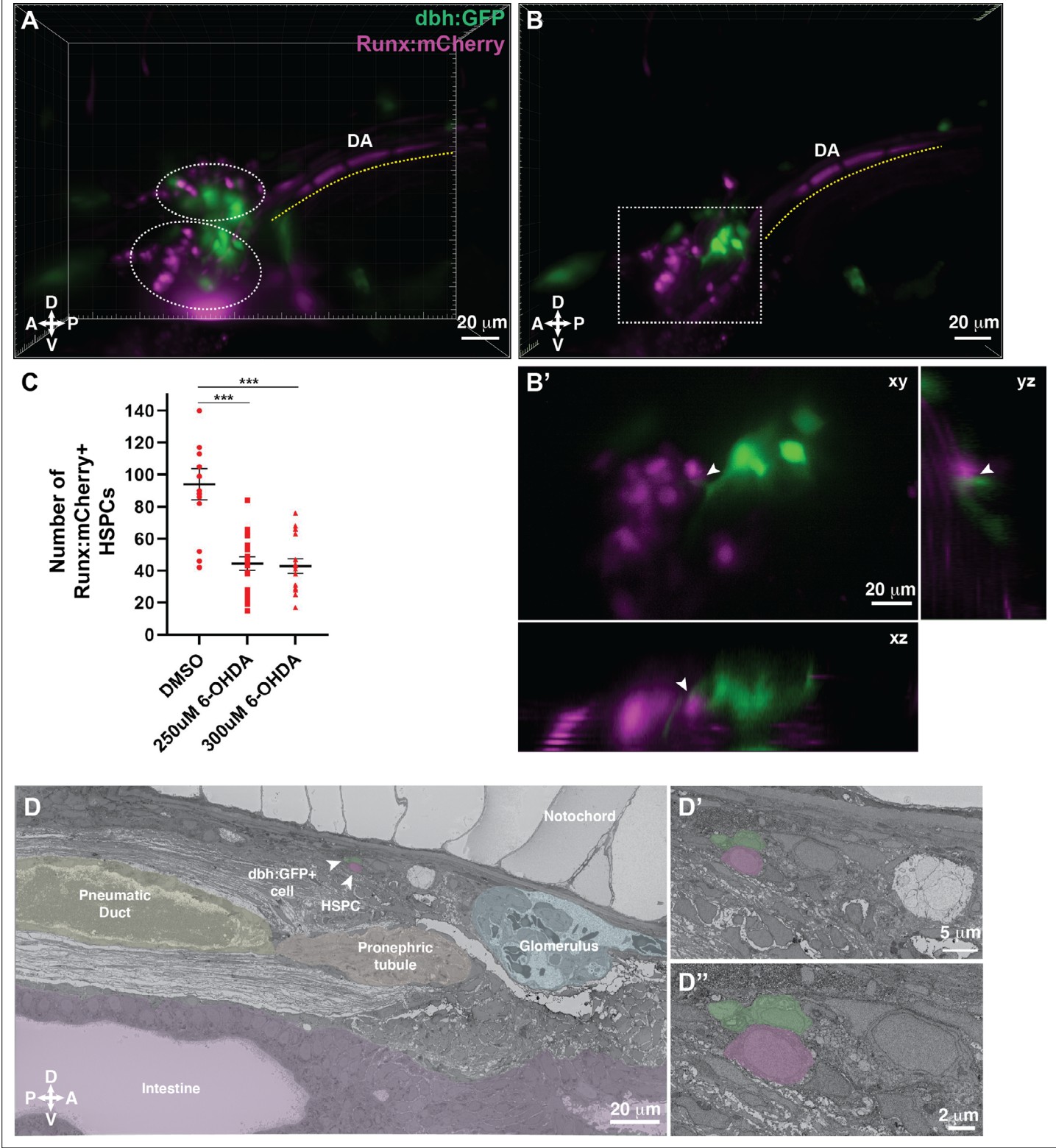

**Figure 7.** Dopamine beta-hydroxylase positive ganglion-like cells are present within the larval kidney marrow niche. (**A**) 3D rendering generated using light sheet movies of Runx:mCherry;dbh:GFP double transgenic larva shows mCherry⁺ clusters (dotted ovals) in close proximity to GFP⁺ cells. DA, dorsal aorta. (**B**) Oblique slice through the 3D volume shows GFP⁺ extensions from the dbh:GFP⁺ cells into the mCherry⁺ hematopoietic stem and progenitor cell (HSPC) clusters. (**B′**) Detail of the boxed region in B shows contact formation between the GFP⁺ extensions and mCherry⁺ HSPCs (white arrowheads) in all three planes. (**C**) Quantification showing significantly reduced number of Runx:mCherry⁺ HSPCs in 6-hydroxydopamine (6-OHDA)-

*Figure 7 continued on next page*

_Figure 7 continued_

treated transgenic larvae compared to DMSO controls. Unpaired t test with Welch's correction. DMSO vs. 250 µM 6-OHDA, p=0.0003; DMSO vs. 300 µM 6-OHDA, p=0.0002. Sample size (n), DMSO, n=13; 250 µM 6-OHDA, n=18; 300 µM 6-OHDA, n=15. Number of biological replicates = 3. (**D**) The ultrastructure of a dbh:GFP+ cell (labeled green) in proximity of a cell with HSPC-like morphology (labeled magenta) (white arrowheads). Surrounding tissues are labeled, and the dbh:GFP+ cell HSPC pair is posterior to the glomerulus, dorsal to pneumatic duct, pronephric tubule and intestine, and ventral to the notochord. (**D′**) Higher magnification shows the dbh:GFP+ cell-HSPC pair is only three-cell diameters from the vessel lumen (white area). (**D″**) Full resolution detail of the dbh:GFP+ cell-HSPC pair showing contact formation. Abbreviations: D, dorsal; V, ventral; A, anterior; P, posterior.

The online version of this article includes the following video and figure supplement(s) for figure 7:

**Figure supplement 1.** Correlative light and electron microscopy (CLEM) confirms that dbh:GFP+ cells are present within the kidney marrow (KM) and neurotoxin targeting dopaminergic cells reduces hematopoietic stem and progenitor cell (HSPC) colonization of the KM niche.

**Figure 7—video 1.** Alignment of confocal and micro-computed tomography (microCT) stacks for the dbh:gfp+ larvae.

https://elifesciences.org/articles/64835/figures#fig7video1

**Figure 7—video 2.** Correlative light and electron microscopy (CLEM) of confocal and serial block-face scanning electron microscopy (SBEM) datasets confirms dbh:gfp+ cells form part of the larval kidney marrow (KM) niche.

https://elifesciences.org/articles/64835/figures#fig7video2

HSPCs in clusters adjacent to the glomerulus, lodged in perivascular regions, or in the lumen of vessels. Many of these perivascular HSPCs were found in pockets of ECs and adjacent or in contact with MSCs, similar to what we previously observed in the CHT (_Tamplin et al., 2015_). HSPCs found in a vessel lumen were in contact with two to three ECS, while those in clusters or the perivascular niche were in contact with three to six ECs (_Table 1_). We found that these HSPC-EC contacts were positive for the tight junction marker ZO-1. Interestingly, _Drosophila_ equivalent occludin junctions have been found to regulate niche interactions with hematopoietic cells and germline stem cells (_Fairchild et al., 2016_; _Khadilkar et al., 2017_). Approximately 50% of putative HSPCs identified in SBEM datasets were in contact with a single MSC, while contact of HSPCs with other putative HSPCs and/or red blood cells was more variable (_Table 1_). Taking all our data together, we only found three examples of putative HSPCs in contact with ganglion-like cells (_Table 1_), and none of these were in HSPC clusters adjacent to the glomerulus. This suggests that there is heterogeneity between sites of HSPC lodgement in the niche, although more definitive conclusions about the degree of heterogeneity will depend on more replicate samples acquired in future studies.

A major outstanding question is if our observed structural heterogeneity between HSPC locations in the niche also indicates functional heterogeneity. This would result in differential regulation of HSPCs depending on the combination of surrounding niche cells. For example, the 1:1 attachment of an HSPC to an MSC that we observe for 50–60% of HSPCs could regulate quiescence vs. expansion. In our previous studies of the CHT, the MSC appeared to provide an anchor for symmetric vs. asymmetric HSPC divisions (_Tamplin et al., 2015_). The CHT of the zebrafish differs from the fetal liver because there is not the massive exponential expansion of HSPCs required to sustain the mammalian embryo in utero (_Kumaravelu et al., 2002_; _Tamplin et al., 2015_). Unlike mammalian embryos, zebrafish embryos develop externally and can survive for many days with no definitive hematopoiesis (_Sood et al., 2010_; _Zhang et al., 2011_). We do not know if the HSPCs that transit through the CHT to the KM have equal potential to become long-term quiescent hematopoietic stem cells, or if there is already heterogeneity within the HSPC pool. Some HSPCs may be fated to produce the blood lineages required during development, while others may retain their quiescence as they transit through the CHT to the KM. Although we do not have the genetic tools yet to functionally distinguish between different sites of HSPC lodgement within the larval KM niche, we are developing a labeling approach that would make this possible. Recent studies showed that subtypes of ECs within the vasculature of the mouse bone marrow microenvironment provide different regulatory functions (_Zhang et al., 2021_). It will be fascinating to unravel how heterogeneity of the HSPC pool interacts with heterogeneity of different sites within the niche, and ultimately decides the functional output of HSPCs.

## Materials and methods

### Animal care

Adult zebrafish (_Danio rerio_) were maintained at 28°C on a 14 hr:10 hr light:dark cycle. The Tg(Runx-1+23:GFP) (_Tamplin et al., 2015_) transgenic line used in this study labels single rare HSPCs, and

Tg(Runx1+23:mCherry) (*Tamplin et al., 2015*) and Tg(cd41:GFP) (*Lin et al., 2005*) label broader HSPC populations. Tg(flk:ZsGreen) (*Cross et al., 2003*) and Tg(flk:mCherry) (*Chi et al., 2008*) mark endothelial cells. Tg(cdh17:EGFP) (*Zhou et al., 2010*) labels the pronephric tubules. Tg(cxcl12/sdf-1a:DsRed2) labels stromal cells (*Glass et al., 2011*). Tg(dbh:EGFP) (*Zhu et al., 2012*) marks sympathetic neurons of superior cervical ganglion. Larval zebrafish were raised in Petri dishes containing E3 solution (5 mM NaCl, 0.17 mM KCl, 0.33 mM $CaCl_2$, and 0.33 mM $MgSO_4$) until 5 dpf. All experiments were performed in accordance with protocols approved by the Institutional Animal Care and Use Committees at the University of Illinois at Chicago (Protocol ACC 19-051) and the University of Wisconsin-Madison (Protocol M006348).

## Plasmid construction

The *mito-APEX2_p2A_APEX2-H2B_p2A_mCherry* cassette was synthesized on a kanamycin-resistant pUC19 backbone (GenScript) and used as a middle entry vector for assembly using LR Clonase II Plus for Multisite Gateway Cloning (Invitrogen). The four plasmids used in the reaction were: (1) pCM293 –6.3 kb*draculin* (*drl*) 5′ entry promoter construct (*Herbomel et al., 1999*; *Mosimann et al., 2015*); (2) synthesized middle entry vector *pME_mito-APEX2_p2A_APEX2-H2B_p2A_mCherry* (a.k.a. *pME-APEX2-mCherry*; Addgene #188944); (3) Tol2kit #302 3′ entry vector *p3E-polyA* (*Kwan et al., 2007*); (4) Tol2kit #394 destination vector *pDestTol2pA2* (*Kwan et al., 2007*). The Gateway assembled LR reaction product, *draculin:mito-APEX2_p2A_APEX2-H2B_p2A_mCherry_polyA* plasmid (a.k.a. *Tol2pA2-drl:APEX2-mCherry*; Addgene #188945), was sequence-verified before injection into one-cell zebrafish embryos.

## Microinjections

Transient transgenesis was established by injecting *Tol2pA2-drl:APEX2-mCherry* plasmid together with *tol2* transposase into wild type zebrafish embryos at the single-cell stage. For light sheet imaging, 1 mg/ml of alpha-bungarotoxin (*Swinburne et al., 2015*) and 2.5% of Oregon Green dye (Dextran, Oregon Green 488; 70,000 MW, Invitrogen) were injected retro-orbitally into 5 dpf transgenic larvae to paralyze them and label their vessels, respectively.

## Whole mount immunofluorescence

The 5 dpf Runx1+23:mCherry[+] zebrafish larvae were fixed overnight in AB buffer containing 4% PFA in ×2 fix buffer (234 mM sucrose, 0.3 mM $CaCl_2$, and ×1 PBS). Post fixation, the larvae were permeabilized with 0.5% Triton in ×1 PBS and washed in deionized water for 2.5 hr. Blocking was performed with 2% BSA in 0.5% Triton/PBS and rabbit polyclonal anti-mCherry primary antibody (Abcam) was used in 1:500 dilution overnight at 4°C. Donkey anti-rabbit Alexa Fluor 568 (Invitrogen) was used as a secondary antibody at 1:1000 dilution. For PH3 labeling, mouse monoclonal anti-PH3 (Ser10, clone 3H10) antibody (Millipore Sigma) was used in 1:250 dilution overnight at 4°C. Donkey anti-mouse Alexa Fluor 647 (Invitrogen) was used as a secondary antibody at 1:1000 dilution.

For ZO-1 antibody staining, the larvae were fixed in 4% PFA overnight followed by 3 PBST (×1 PBS with 0.1% Tween 20) rinses. Larvae were dehydrated in increasing methanol/PBST series (25%, 50%, 75%) and stored in 100% methanol at –20°C overnight followed by rehydration in the reversed methanol/PBST series (75%, 50%, 25%). After PBST rinses, they were blocked in 2% serum solution in PBDT (×1 PBS with 1% BSA, 1% DMSO, and 0.5% Triton) and then incubated in mouse monoclonal ZO-1 (clone 1A12) primary antibody (Thermo Fisher Scientific) in 1:500 dilution overnight at 4°C. Donkey anti-mouse Alexa Fluor 647 (Invitrogen) was used as a secondary antibody at 1:1000 dilution.

## Drug treatment

Runx1+23:mCherry[+] zebrafish larvae were treated at 4 dpf with 250 and 300 µM 6-OHDA (Cayman Chemical) for 24 hr. The drug was washed off at 5 dpf followed by fixation, tissue clearing, and imaging.

## Tissue clearing and confocal imaging of KM

To image the KM in fixed samples, following antibody staining, the larvae were embedded in 1% low melting agarose in glass-bottomed dishes (35 mm, MatTek Corporation) with the dorsal surface in contact with the glass bottom. To improve visibility through the larvae, optical tissue clearing

was performed according to the protocol (*Dodt et al., 2007*) with some modifications. Embedded samples were washed six to seven times with 100% methanol, followed by overnight incubation at 4°C to dehydrate the tissues. Next, the samples were washed six to seven times with a 1:1 ratio of 100% methanol and BABB clearing reagent (1 part benzyl alcohol:2 parts benzyl benzoate) and incubated for 30 min at room temperature before transferring to 100% BABB. These organic solvents have high refractive indices which result in the solvation of lipids within tissues to align their refractive indices to prevent scattering of light through the tissues (*Dodt et al., 2007*). Zeiss LSM 880 confocal microscope was used to image the KM. Volume dimensions of $354.25 \times 354.25 \times 74 \ \mu m^3$ was acquired every 2 µm $(0.35 \times 0.35 \times 2 \ \mu m^3/\text{pixel})$ with a line sequential scan mode and averaging of 2.

For ZO-1 antibody staining, the stained larvae were sectioned in the sagittal plane through vibratome sectioning. Sections of 100 µm were generated that were imaged using Olympus FluoView 1000 at ×60 magnification.

## Light sheet imaging

For light sheet imaging, stage-matched transgenic zebrafish larvae were paralyzed by retro-orbital injection of alpha-bungarotoxin (*Swinburne et al., 2015*) and embedded in 1% low melting point agarose within thin capillary tubes. The KM was illuminated with a light sheet from one axis, and fluorescence was detected at a perpendicular axis. Larvae were imaged with a ×20 objective using the Zeiss Light sheet Z.1 for Single Plane Illumination Microscopy (SPIM). The capillary tube was inserted into the sample chamber to release the larva into the chamber containing E3 while remaining attached to the capillary tube through the agarose layer. The larva was rotated such that the lateral surface of the larva closer to the edge of the agarose layer was perpendicular to the detection objective to optimize fluorescence detection. The laser path was aligned to illuminate the KM from one axis. Z stacks were acquired through the entire KM with 1 µm spacing in less than 1 min and time-lapse videos were recorded to track the circulating HSPCs entering the KM. Through a full 360° rotation of the larva relative to the detection lens, only about 5° were optimal for light emission from the sample and image acquisition of the KM. Some additional light sheet images were acquired using a Mizar Tilt microscope.

## Image analysis

Image processing and rendering were done using Imaris.v.9 (Bitplane), IMOD (*Kremer et al., 1996*), and ImageJ/Fiji (*Schindelin et al., 2012*). Imaris was used to count the number of cells within the KM niche. Imaris 'Spot' module was used first to specify the ROI around the KM in confocal images followed by automated counting of the spots generated corresponding to the single HSPCs. The accuracy of the spots generated was further confirmed by using the manual mode of 'Spot' function. 'Co-localization' feature on Imaris was used to identify PH3/mCherry double positive cells. Distances between HSPCs and niche cells were measured in 3D light sheet image volumes using Imaris. In microCT datasets, ImageJ/Fiji was used to orient and define the ROI that was trimmed and then scanned by SBEM. The 'Add image to' function in Imaris was used to add a light sheet or confocal dataset to an already open SBEM dataset, followed by alignment of both datasets in three dimensions, first using gross anatomical markers (e.g., somites, vessels), then single cells and/or nuclei as reference. Orthogonal views through the two datasets in the XY, YZ, and XZ planes were used to confirm the alignment. After alignment of light microscopy and SBEM datasets, single putative HSPCs and their surrounding niche cells were manually traced in high-resolution SBEM datasets using IMOD software. Separate objects were defined by drawing contours around the plasma membrane of the target cell (and sometimes nuclei) as it moved slice by slice in the Z plane. Individual contours were meshed with *imodmesh* to reveal 3D reconstructions of the cells of interest. In this way, HSPCs, ECs, MSCs, GLs, and their nuclei were identified and represented in the 3D model. Fluorescent image levels and background were adjusted, using the default 'background subtraction' (Imaris), threshold adjustment, and/or brightness/contrast adjustment (Fiji/ImageJ, Imaris, Zeiss Zen, Adobe Photoshop). Imaris was used to define 'Spots' in the cd41:GFP+ CLEM dataset using default settings, except 'Spot Detection' was set to: Estimated XY diameter: 3.00 µm; model PSF-elongation along Z axis 'ON', estimated Z diameter: 6.00 µm; background subtraction 'ON'. SBEM brightness/contrast adjustment was performed using IMOD and/or Fiji, Imaris, Adobe Photoshop.

## Larvae preparation for APEX2$^+$;mCherry$^+$ CLEM (Workflow #1)

Five dpf zebrafish larvae were prepared for microCT and SBEM as previously described (*Deerinck et al., 2010*). Briefly, immediately after light sheet imaging, larvae were fixed in 2.5% glutaraldehyde and 4% paraformaldehyde in 0.15 M cacodylate buffer (CB, pH 7.4) at 4°C overnight. After removing the fixative, larvae were treated for 15 min with 20 mM glycine in 0.15 M CB, on ice to quench the unreacted glutaraldehyde. Preincubation was done in 2.5 mM DAB solution (25.24 mM stock in 0.1 M HCl) in 0.15 M CB for 1 hr on ice. For staining, 0.03% $H_2O_2$ containing DAB solution was added to the larvae on ice. The reaction was monitored every 5–10 min and stopped when the desired intensity was achieved. The staining buffer was washed off with ×5 washes using 0.15 M CB. Then, larvae were washed with 0.15 M CB and then placed into 2% $OsO_4$/1.5% potassium ferrocyanide in 0.15 M CB containing 2 mM $CaCl_2$. The larvae were left for 30 min on ice and then 30 min at room temperature (RT). After thorough washing in double distilled water (ddH$_2$O), larvae were placed into 0.05% thio-carbohydrazide for 30 min. Larvae were again washed and then stained with 2% aqueous $OsO_4$ for 30 min. Larvae were washed and then placed into 2% aqueous uranyl acetate overnight at 4°C. Larvae were washed with ddH$_2$O at RT and then stained with 0.05% en bloc lead aspartate for 30 min at 60°C. Larvae were washed with ddH$_2$O and then dehydrated on ice in 50%, 70%, 90%, 100%, 100% ethanol solutions for 10 min at each step. Larvae were then washed twice with dry acetone and placed into 50:50 Durcupan ACM:acetone overnight. Larvae were transferred to 100% Durcupan resin overnight. Larvae were then flat embedded between glass slides coated with mould-release compound and left in an oven at 60°C for 72 hr.

## Larvae preparation for GFP$^+$ CLEM (Workflow #2)

To facilitate the 3D correlation between different imaging modalities when using any GFP$^+$ transgenic line, we added DRAQ5, a nuclear DNA binding fluorescent dye. Briefly, the 5 dpf transgenic zebrafish larvae were fixed with 0.5% glutaraldehyde and 2% paraformaldehyde in 0.15 M cacodylate buffer (CB, pH 7.4) for 2.5 hr and 100 µm thick sagittal sections were collected and incubated in DRAQ5 (1:1000, Cell Signaling Technology) on ice for an hour. Confocal images of GFP and DRAQ5 signals were collected on a Leica SPE II confocal microscope with a 20x oil-immersion objective lens using 488 and 633 nm excitation. Following, the samples were preincubated in 2.5 mM DAB solution (25.24 mM stock in 0.1 M HCl) in 0.15 M CB for 30 min on ice. Next, the photo-oxidation of DAB by DRAQ5 was done in 2.5 mM DAB in 0.15 M CB using a solar simulator (Spectra-Physics 92191-1000 solar simulator with 1600 W mercury arc lamp and two Spectra-Physics SP66239-3767 dichroic mirrors to remove infrared and ultraviolet wavelengths), while bubbling oxygen in the solution. The light was filtered through a 10 cm square bandpass filters (Chroma Technology Corp.) for illumination at 615 nm (40 nm band pass). The reaction was monitored every 20 min and stopped when the desired darkening in the nuclei was achieved. Larvae were then washed five times with 0.15 M CB and incubated in 2% $OsO_4$/1.5% potassium ferrocyanide in 0.15 M CB containing 2 mM $CaCl_2$ to get an EM visible stain. The larvae were left for 30 min on ice and then 30 min at RT. After thorough washing in ddH$_2$O, larvae were placed into 0.05% thiocarbohydrazide for 30 min. Larvae were again washed and then stained with 2% aqueous $OsO_4$ for 30 min. Larvae were washed and then placed into 2% aqueous uranyl acetate overnight at 4°C. Larvae were washed with ddH$_2$O at RT and then stained with 0.05% en bloc lead aspartate for 30 min at 60°C. Larvae were washed with ddH$_2$O and then dehydrated on ice in 50%, 70%, 90%, 100%, 100% ethanol solutions for 10 min at each step. Larvae were then washed twice with dry acetone and placed into 50:50 Durcupan ACM:acetone overnight. Larvae were transferred to 100% Durcupan resin overnight. Larvae were then flat embedded between glass slides coated with mould-release compound and left in an oven at 60°C for 72 hr.

## MicroCT and SBEM

The microCT tilt series were collected using a Zeiss Xradia 510 Versa (Zeiss X-Ray Microscopy) operated at 80 kV (88 µA current) with a 20x magnification and 0.872 µm pixel size. MicroCT volumes were generated from a tilt series of 2401 projections using XMReconstructor (Xradia). SBEM were accomplished using Merlin SEM (Zeiss, Oberkochen, Germany) equipped with a Gatan 3View system and a focal nitrogen gas injection setup (*Deerinck et al., 2018*). This system allowed the application of nitrogen gas precisely over the blockface of ROI during imaging with high vacuum to maximize the SEM image resolution. Even a minimal accumulation of charge on the specimen surface resulted

in poor image quality and distortions in the non-conductive biological samples. This could happen through 3000 sequential images of the charge-prone zebrafish blockface due to image jitter. Deerinck et al. recently introduced a new approach to SBEM that uses focal nitrogen gas injection over the sample block surface to eliminate the charging, while allowing the high vacuum to be maintained in the specimen chamber, resulting in a tremendous improvement in image resolution, even with specimens that were not intensely heavy-metal stained (*Deerinck et al., 2018*). Images were acquired in 3 kV accelerating voltage and 0.5 µs dwell time; Z step size was 70 nm; raster size was 30k × 18k and Z dimension was ~3000 image samples. Volumes were collected using 40% nitrogen gas injection to samples under high vacuum. Once volumes were collected, the histograms for the slices throughout the volume stack were normalized to correct for drift in image intensity during acquisition. Digital micrograph files (.dm4) were normalized and then converted to MRC format. The stacks were converted to eight bit and volumes were manually traced for reconstruction using IMOD (*Kremer et al., 1996*).

## Acknowledgements

This work was supported by grants from the NIH NHLBI (R01HL142998), NIDDK (K01DK103908), American Society of Hematology (Junior Faculty Scholar Award; OJT), and the American Heart Association (Grant #19POST34380221; SA). The NCMIR is principally supported by grants from the NIH NINDS (1U24NS120055-01) and NIGMS (R24 GM137200). Light sheet imaging was performed at the Integrated Light Microscopy Core Facility at the University of Chicago with the help of Christine Labno and Vytas Bindokas, and at the UW-Madison Optical Imaging Core with Lance Rodenkirch. The authors wish to thank: Dr Daniela Boassa for her assistance in EM probe design with APEX2; Tom Deerinck, Steven Peltier, and Tristan Shone for technical advice with Focal CC; Iain A Drummond for feature identification in SBEM sections; Christian Mosimann for the draculin promoter 5' entry vector; Willy Wong for depositing the SBEM datasets in The Cell Image Library. Some of this work was performed at the University of Illinois at Chicago, Chicago, United States.

## Additional information

### Competing interests

Owen J Tamplin: Reviewing editor, eLife. The other authors declare that no competing interests exist.

### Funding

| Funder | Grant reference number | Author |
| --- | --- | --- |
| National Heart, Lung, and Blood Institute | R01HL142998 | Owen J Tamplin |
| National Institute of Diabetes and Digestive and Kidney Diseases | K01DK103908 | Owen J Tamplin |
| American Heart Association | 19POST34380221 | Sobhika Agarwala |
| National Institute of Neurological Disorders and Stroke | 1U24NS120055-01 | Mark H Ellisman |
| National Institute of General Medical Sciences | R24 GM137200 | Mark H Ellisman |
| American Society of Hematology | Junior Faculty Scholar Award | Owen J Tamplin |

The funders had no role in study design, data collection and interpretation, or the decision to submit the work for publication.

## Author contributions
Sobhika Agarwala, Data curation, Formal analysis, Validation, Investigation, Visualization, Methodology, Writing – original draft, Writing – review and editing; Keun-Young Kim, Data curation, Formal analysis, Validation, Investigation, Visualization, Methodology, Writing – review and editing; Sebastien Phan, Software, Formal analysis, Visualization; Saeyeon Ju, Ye Eun Kong, Guillaume A Castillon, Formal analysis; Eric A Bushong, Investigation, Methodology; Mark H Ellisman, Conceptualization, Resources, Supervision, Funding acquisition, Methodology, Project administration; Owen J Tamplin, Conceptualization, Resources, Formal analysis, Supervision, Funding acquisition, Validation, Investigation, Visualization, Methodology, Writing – original draft, Project administration, Writing – review and editing

## Author ORCIDs
Eric A Bushong ⓘ https://orcid.org/0000-0001-6195-2433
Owen J Tamplin ⓘ https://orcid.org/0000-0001-9146-4860

## Ethics
All experiments were performed in accordance with protocols approved by the Institutional Animal Care and Use Committees at the University of Illinois at Chicago (Protocol ACC 19-051) and the University of Wisconsin-Madison (Protocol M006348).

## Decision letter and Author response
Decision letter https://doi.org/10.7554/eLife.64835.sa1
Author response https://doi.org/10.7554/eLife.64835.sa2

---

# Additional files

## Supplementary files
MDAR checklist

## Data availability
SBEM datasets have been deposited in the National Center for Microscopy and Imaging Research (NCMIR) publicly accessible resource database Cell Image Library (CIL). There are six SBEM datasets (accession numbers: CIL:54845, CIL:54846, CIL:54847, CIL:54848, CIL:54849, CIL:54850) that are accessible as group with the following link: http://cellimagelibrary.org/groups/54850. CIL accession numbers are referenced in Table 1. Newly generated plasmids have been deposited in Addgene (#188944 and #188945).

The following datasets were generated:

| Author(s) | Year | Dataset title | Dataset URL | Database and Identifier |
|---|---|---|---|---|
| Kim KY, Agarwala S, Phan S, Ju S, Kong YE, Castillon GA, Bushong EA, Ellisman MH, Tamplin OJ | 2022 | Transverse sections of 5 days post-fertilization wild-type zebrafish larva in the region of the anterior kidney | https://doi.org/10.7295/W9CIL54845 | CIL:54845, 10.7295/W9CIL54845 |
| Kim KY, Agarwala S, Phan S, Ju S, Kong YE, Castillon GA, Bushong EA, Ellisman MH, Tamplin OJ | 2022 | Sagittal sections of 5 days post-fertilization drl:APEX2-mCherry+ zebrafish larva in the region of the anterior kidney (eLife Table 1: APEX2 #2 5 dpf) | https://doi.org/10.7295/W9CIL54846 | CIL:54846, 10.7295/W9CIL54846 |
| Kim KY, Agarwala S, Phan S, Ju S, Kong YE, Castillon GA, Bushong EA, Ellisman MH, Tamplin OJ | 2022 | Sagittal sections of 5 days post-fertilization drl:APEX2-mCherry+ zebrafish larva in the region of the anterior kidney (eLife Table 1: APEX2 #1 5 dpf) | https://doi.org/10.7295/W9CIL54847 | CIL:54847, 10.7295/W9CIL54847 |

*Continued on next page*

*Continued*

| Author(s) | Year | Dataset title | Dataset URL | Database and Identifier |
|---|---|---|---|---|
| Kim KY, Agarwala S, Phan S, Ju S, Kong YE, Castillon GA, Bushong EA, Ellisman MH, Tamplin OJ | 2022 | Sagittal sections of 5 days post-fertilization dbh:gfp+ zebrafish larva in the region of the anterior kidney (eLife Table 1: dbh:GFP 5 dpf) | https://doi.org/10.7295/W9CIL54848 | CIL:54848, 10.7295/W9CIL54848 |
| Kim KY, Agarwala S, Phan S, Ju S, Kong YE, Castillon GA, Bushong EA, Ellisman MH, Tamplin OJ | 2022 | Sagittal sections of 5 days post-fertilization cd41:gfp+ zebrafish larva in the region of the anterior kidney (eLife Table 1: cd41:GFP 5 dpf) | https://doi.org/10.7295/W9CIL54849 | CIL:54849, 10.7295/W9CIL54849 |
| Kim KY, Agarwala S, Phan S, Ju S, Kong YE, Castillon GA, Bushong EA, Ellisman MH, Tamplin OJ | 2022 | Transverse sections of 5 days post-fertilization wild-type zebrafish larva in the region of the anterior kidney | https://doi.org/10.7295/W9CIL54850 | CIL:54850, 10.7295/W9CIL54850 |

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
