## [Editor Report]

The manuscript reports on an extensive body of work, achieving the still highly challenging identification of HSPCs within the ultrastructure of their niche. The study highlights the heterogeneous nature of HSC-niche interactions, which is consistent with heterogeneity identified through genomic and functional studies. The work presented is of high interest to the field.

---

## [Decision Letter]

**Decision letter after peer review:**

Thank you for submitting your article "Defining the ultrastructure of the hematopoietic stem cell niche by correlative light and electron microscopy" for consideration by *eLife*. Your article has been reviewed by 2 peer reviewers, and the evaluation has been overseen by a Reviewing Editor and Didier Stainier as the Senior Editor. The following individuals involved in review of your submission have agreed to reveal their identity: Anne Schmidt (Reviewer #3).

Three reviewers (including myself) have now carefully read and considered your manuscript, and we all agree that it presents a very interesting study and a technical tour de force establishing multimodal imaging of HSPC niches in the zebrafish kidney, nicely combining confocal microscopy, light sheet microscopy and serial block-face scanning electron microscopy. The results achieved are interesting but still overall preliminary and we agree that major revisions should be completed prior to final publication.

Essential revisions:

The main points of agreement are the following:

1) Multiple different lines of fish are used to highlight HSPCs, but since their signals do not overlap, it is hard to come to a definitive conclusion of what cells are analysed each time. It seems that Runx1:GFP would be the best line to use, but it is actually the least used. CD41:GFP is used in some analyses too and especially in the validation of the APEX transgenic line (though not shown).

2) Only two examples of HSCPs surrounded by the 7 cells niche are presented. Is this the case for all HSPCs, or what sort of variation is observed? Variation would be expected given the reporters used are marking HSPCs rather than primitive HSCs, and therefore a heterogeneous population. In general, more data should be presented and statistically analysed to support the conclusions presented.

3) APEX signal is not always shown, but should be used to unequivocally identify the cell presented.

4) In figure 2 ZO-1 does not look particularly enriched around Cherry positive cells, and the relative conclusions should be rephrased.

5) Scale bars are not always clear, and a number of figure panels are hard to read (see specific comments). The same applies to multiple videos.

6) One point to be better discussed is how the Authors reconcile the similar organisation of CHT and KM niches, when functionally they are very different, ie they support rapidly expanding and more quiescent HSPCs, respectively.

7) Is it the case that there are two anatomically and functionally distinct niches in the KM, the glomerular and perivascular one? The functional difference needs to be proven. One possibility would be to identify differential influence of dopamine on these two niches.*Reviewer #1 (Recommendations for the authors):*

ED1A. Could some contrast indicating the glomerulous and tubules be shown?

ED1I. Please show average and error bars rather than the summarised pie chart.

Video 1 is very hard to follow as cells and structures move significantly from one frame to the next.

Video 2. Perhaps the more novel finding is that most of the interactions are relatively short lived. This should be at least noted and commented on.

Figure 2C. Why CD41gfp and not Runx1:GFP? DsRed resolution is not sufficient to rule out that each HSPC is in touch with just one MSC.

Line 106. Most other HSPCs were within 1 cell diameter: 4 vs 8 are further.

The characterization of the new transgenic fish line needs to be more precise and better shown. How can there be 1 or 2 HSPC/EC pockets, but many more HSPC clusters? IS there a functional difference between the single cells in EC pockets and the cells in clusters?

It is explained that the APEX-cherry cells are identified as also GFP+ in CD41+ fish (why not Runx-GFP?), but there is no CD41-GFP in Figure 3. Are the APEX-cherry cells truly HSPCs?

Figure 5. How many HSPCs are observed in this kind of niche in how many fish? The reporters used are not very strict for primitive HSPCs and variability would be expected and should be shown. At the moment it cannot be claimed that 'HSPCs lodge in a highly ordered multicellular niche'.

Figure 7 and 8. APEX staining is missing but CLEM is equally completed. Is APEX necessary then?

*Reviewer #2 (Recommendations for the authors):*

The manuscript by Agarwala S et al., is a very nice and interesting study and a technical tour de force to establish CLEM as an approach to study the stem cell niche composition. The authors use a variety of genetic reporters in zebrafish and multimodal imaging, comprising confocal and light sheet microscopy, microCT and serial block-face scanning electron microscopy (SBEM) to study the initial stages of hematopoiesis in the developing kidney marrow. The results suggest that the approach is "doable" and can provide qualitative and descriptive information, therefore significantly adding to the field. Among the initial description provided by this technology is the identification of several cell types that might have niche function for hematopoietic stem cells (HSCs) in the kidney marrow, including endothelial cells, mesenchymal stromal cells (MSCs), sympathetic neurons and glial cells, resembling the results obtained by other groups in the mouse bone marrow. However, the data provided is essentially very nice images of individual HSCs or few cells that might well be representative examples, but a systematic study providing quantitative information and analysis to support the conclusions seems to be lacking. It is not clear how many zebrafish have been used in each experiment to obtain conclusions, as many images focus on a single HSPC. In fact, n numbers are only indicated in Figure 8, where 3 biological replicates have been studies. The conclusions would require additional experiments and statistical analysis of the results. Other comments are below:

1) The authors indicate that Runx:GFP is a better HSPC marker than Runx:mCherry. Yet the data with Runx:GFP are scarce. Runx:GFP is only used in Figure 1 and the less specific Runx:mCherry is used in the other main figures. Figure 1 also lacks quantification. How many HSPCs are surrounded by how many endothelial cells? Where are GFP+ cells located in relationship to other niche cells?

2) The authors provide two examples of HSPCs surrounded by 7 cell niche (5 ECs, 1 MSC, 1 Glial like cell), but we don't know if this the case for every HSPC or just a subset/what proportion.

3) Figure 7 importantly identifies putative HSPCs based on morphological criteria, but again only few examples are shown and confirmation that these are HSPCs using the reporter lines seems required.

4) In Figure 8D, HSPCs have been identified as 'morphologically resembling an HSPC'; some of the reporters available should be used to demonstrate that this is the case.

5) L. 101: "This suggests that ZO-1 is a potential candidate for mediating adhesion between HSPCs and the surrounding niche cells in the KM niche" and L. 244-245 of Discussion – Figure 2A does not show a particular enrichment of ZO-1 expression near mCherry+ cells.

6) L.226-7: "suggesting a role for dopamine signalling in colonization of the presumptive adult KM niche" should probably be rephrased. There could be different reasons and neurotransmitters/other molecules responsible for this effect.

7) Scale bars in some figures e.g 3D-G weren't always clear.*Reviewer #3 (Recommendations for the authors):*

The work presented by Owen Tamplin and collaborators is aimed at defining the different cellular components and chemical signals that provide structural and functional specificity to the niche into which hematopoietic stem cells and progenitors (HSPCs) are homing. These investigations are essential to determine precisely the niches that will favour the survival and maintenance of hematopoietic stem cells with the highest degree of stemness which, ultimately, should allow producing this type of cells in the context of regenerative medicine.

Importantly, this kind of work would have not been possible in any other vertebrates, which is why the developing zebrafish is a unique living model organism, owing to its transparency and relative easiness for genetic engineering.

Here, Owen Tamplin and collaborators succeed in providing impressive 3-dimensional information on the structural organization of pre-definitive niches in the developing kidney of the zebrafish larva. In analogy with what has been described in the mammalian bone marrow, they also provide functional evidence for a role of dopamine signalling via glial-like cells, which empowers the use of the zebrafish to comprehend the biology of immunity establishment, a process essential for life. Overall, this piece of work paves the way for further, more detailed analyses on the physical and fonctional interactions between hematopoietic stem cells and their 3D-niche environment.

Comments to the Authors

In this manuscript, the authors address, at high spatial 3-dimensional resolution that includes light sheet microscopy on the whole zebrafish larva and correlative light/electron microscopy, critical aspects of hematopoietic stem cells and progenitors (HSPCS) and their niches: their conserved structural and cellular identities as well as the regulatory function of contacting glial-like cells.

The work is overall quite impressive, ambitious technically, and provides unprecedented 3D reconstitution of the pre-definitive hematopoietic niche region, in the developmental zebrafish kidney (the future functional equivalent of the adult bone marrow in mammals). However, I think that the manuscript, in its present form, requires additional experimental work, in particular regarding physical contact between HSPCs (unambiguously identified) and glial-like cells (see comments on Figure 8 and text), which is the functional piece of work of the study. In addition, the manuscript is difficult to follow for the reader in particular for the figures that are, for many of them, lacking annotations (even with the support of Extended Data, the reader has to make serious efforts to be able to critically analyze the Figures (which is clearly a difficult task for authors that are publishing correlative images and EM analysis)).

Finally and conceptually, it would be good if the authors could propose a more precise vision in regard to the potential functional implications of the 2 geographical/structurally defined niches described here, i.e the one proximal to the glomerulus, and the more distant perivascular niche (this should be, on my opinion, included in the Discussion of the manuscript). Somehow, along the manuscript, we lose the track on the authors aim regarding the characterization of these 2 niche types/regions respective peculiarities (number of HSPCs versus other cell types, influence of neurotransmitters, etc …) and, in fine, the perivascular niche in which 'rare' cells are homing seems to be the centre of attention without any further and precise explanation.

Please find my major comments beneath on Figures and related text:

– Figure 3 and text

The information is not easily captured by the reader.

In general, panels D-G need annotations (I would suggest: DA and ISVs in D, the position of DAB stained regions in E, the main organs that we see in F (I guess we see the notochord and part of the swim bladder?), G: the position of somites (s1, s2, s3, s4; with somite 2 nearby the kidney marrow as shown in the very nice suppl. Video 4)).

More specific comments on specific panels:

D: I doubt that the large, pink, elongated structure is indeed HSPCs (could it be the auto-fluorescence (AF) from the gut as seen in Extended Data Figure 3? I guess this is the same for Figure 4, panel A)

E: the local DAB precipitate(s) is not clearly visible (could the authors make attempt to obtain an image before and after DAB staining, on the same larva (and show the comparison of the 2 of them?)).

For the accompanying Extended Data Figure 5, the legend of panel D says we see a reconstitution of HSPCs and multicellular niche (line 851), where are they? could the authors delineate the corresponding regions?

For the Supplementary Video 5, which is truly impressive, and beautiful in resolution, with plenty of information, what a pity not to have annotations to get a maximum benefit from it; the frustration is immense ! I propose that authors decompose it for the images that are the most in relation to HSPC clusters and the relevant surrounding organs (including vascular structures) and build an additional piece of Extended Data. I realize that one of the sections is composing Figure 5A (00:15 of the pile of images), which should appear in the legend to this Figure. In addition, are the panels Bi-iii of Figure 5 also extracted from Supplementary Video 5 ? In addition, for Video 5, should we expect to see the two different regions in which clusters and single (or 1-2) cells are supposed to establish 2 specific (and different) niches?

Finally, it appears that red blood cells (RBCs, which are nucleated in the zebrafish) have a very dark cytoplasm 00:11 for ex ? is it DAB staining (in the cytoplasm then and not in nuclei?) or is it endogenous cytosolic peroxydase activity which is revealed here?

In the text that relates to Figure 3, I find a potential inconsistency in the intention of having sparse labelling of HSPCs (line 131), which I understand is an obvious advantage if one wants to increase the chance to visualize single cells with no other labelled contacting hematopoietic cell (and hence using F0 larvae), but to have as an outcome the same result as when using stable transgenic line (see lines 139-140) is rather unexpected. Could the authors comment on that?

Also, do the 1-2 mCherry+ cells/EC pockets and 1-5 mCherry+/clusters were observed in each of the 11 larvae that were analyzed (owing to expected mosaicism, this may not be the case)?

– Figure 4 and text

Figure 4 is quite informative for comprehending the strategy followed technically but, somehow, is not illustrating the text lines 173-175 that makes the statement '… dark nuclear and mitochondrial staining … confirming that we have identified the same APEX2+/mCherry+ HSPC …'; thus, the authors should provide a magnification of that HSPC (from the EM image in panel Dii, with a clear nuclear and mitochondrial dark APEX2-derived staining).

– Figure 5 and text

I find an inconsistency saying that the posterior perivascular niche encloses a single HSPC as stated line 187 (or may be 2 maximum? as stated lines 137 and 140 if I understand correctly, which is also considered as a 'single rare' cell in Figure 6) (with the ratio 1xHSPC/5xEC/1xMSC) when the figure (see also the legend of Biii lines 709-710) shows two additional – non labelled – HSPCs (HSPCs 2 and 3); can the authors comment on that?

Supplementary Video 6 is impressive, but again (in particular for the planes 00:00 to 00:33), some annotations should be added on most relevant tissues/organs/cell types.

In addition, and on conceptual grounds, it is proposed that the CHT niche(s) is (are), in their vast majority, the site(s) of progenitors expansion, meaning most probably a very minor proportion would home cells bearing/maintaining full, long-term HSC potential (as opposed to the kidney marrow in which future adult HSCs are homing, thus with an expected higher proportion of HSC-specific niches). Based on this, how would the authors conceive that such conserved structures would accommodate such differentially fated cells (see the statement lines 187-188 'demonstrating this cellular structure is conserved between developmental and presumptive adult niche')?

– Figure 8 and text

The Figure with the light sheet information on close proximity between HSPCs and dbh cells is nice and the quantitative analysis panel (C) convincing.

However, the data should be substantiated using a double Tg line (draculin:2APEX2-mCherry) X (dbh:GFP) to confirm the identity of the HSPC(s) contacting the dbh cell(s) and not only a resemblance as state line 231 (hence labelled by the APEX2 nuclear + mitochondrial activities). This would very much substantiate the work and strengthen the manuscript on this important aspect of HSPC/niche establishment/regulation (and the authors have all the tools/approaches at hand). This is also required since the authors clearly state at the end of the Discussion (line 262) '… additional rounds of CLEM to confirm that dbh:GFP+ cells were adjacent to HSPCs …'; clearly, identifying unambiguously HSPCs is mandatory.

In the accompanying Extended Data Figure 9, panel (A) line 892, what do (n=10) refers to? 10 sections of 1 experiment? 1 section, each from 10 independent experiments? What cells does the blue colour in panel Bii underline?

In the text, the authors say that they wish to address a role of dopamine signalling in the KM niche colonization (hence they treat with 6-OHDA from 4 to 5dpf, during niche colonization); would a shorter treatment (ex: few hours) also affect HSPCs in the niche, which would indicate a function of dopamine in the niche per se (ex: survival, maintenance) rather than colonization?

Finally, the authors propose, if I understand correctly, that there may be 2 different types of functional niches (1) proximal/adjacent to the glomerulus, (2) more distant and referred to as posterior perivascular niche; do they have any information on a possible differential influence of dopamine and dbh-cells on these 2 niche types (or to their colonization)? Answering to that question would as well strengthen the functional impact of the work.

Comments on the Discussion:

The authors should, in my opinion, discuss the possible functional discrepancies between the 2 geographic positions of the niches that are visualized here, i.e the one adjacent to the glomerulus and the one that is perivascular (more distant from the glomerulus); they should do so also in the context of the current work on mammalian bone marrow niches. This is an important issue in the field because, ultimately, one wants to comprehend what are the functional differences between the niches that allow the maintenance of long-term HSCs and the ones that are more devoted to support specific lineage differentiation. Do the authors have a vision on that issue?

[Editors’ note: further revisions were suggested prior to acceptance, as described below.]

Thank you for resubmitting your work entitled "Defining the ultrastructure of the hematopoietic stem cell niche by correlative light and electron microscopy" for further consideration by *eLife*. Your revised article has been evaluated by Didier Stainier (Senior Editor) and a Reviewing Editor.

The manuscript has been improved and all reviewers commended the extra work and data added to the study. They also agree that it is important for the readers to include a few further edits to better reflect the limitations of the techniques used and the heterogeneity uncovered when a larger number of HSPC is analysed. However, no further experiments are needed. Please see below for details of their recommendations.

*Reviewer #2 (Recommendations for the authors):*

The manuscript describing this technically groundbreaking and very impressive approach to study the niche is largely improved and provides a more balanced interpretation of the results, considering the inherent limitation of the approach including datasets available.

I understand that the approach is technically very challenging and time-consuming, and would not allow to obtain larger datasets to perform quantitative analyses and obtain more definitive conclusions. Therefore, these should be rephrased to emphasise the qualitative (rather than quantitative) nature of the analysis and acknowledge some limitations (such as the lack of markers for putative HSCs in some cases). The contact with glial cells is emphasised but it should be noted that only 3 out of 22 HSPCs were found in contact with glial cells. Besides the numerical difference of HSPCs found in contact with different niche cells, whether HSPCs contacting these different niche cells might be functionally distinct could be incorporated into the discussion. The conclusions should acknowledge the qualitative nature of the study and emphasise the strengths and limitations of this novel approach to study the niche in the future, since it could potentially elicit significant interest in the field to use it in different experimental settings.

*Reviewer #3 (Recommendations for the authors):*

Many thanks to the authors to have provided a significant revision of their work; obviously, this correlative multimodal microscopy work is a tedious, difficult, time-consuming but essential task.

While many of my points have been taken into account, there still remains 2 critical aspects that I believe should be addressed and that are partly emphasized by the new dataset (dataset 4 using the CD41:gfp line, workflow 2).

Point 1 (that also relates to essential point 2 of the Evaluation Summary): The authors have provided a new dataset (see new Figure 6 and the 2 Supplemental Figures+ Table 1) that significantly increases the number of HSPCs that are analysed, including the identification of potential niche contacting cells (Table 1). However, the dataset raises the critical point of heterogeneity of the niche environment because the cluster of HSPCs that is analysed appears disconnected to any glial-like cells (GL, see outcome = 0 in Table 1 for the 17 HSPCs). Hence, this new dataset strengthens the idea of the heterogeneity which is not quantitatively appreciated in the work. Also, on qualitative aspects, it appears that the HSPC cluster is more distal to the glomeruli than the HSPCs illustrated in Figure 5 and Figure 5 – figure supplement 1 and 3 for example (which appear from the images to be more proximal to the glomerulus while for new Figure 6 the cluster appears to be connecting the pronephric tubule and the margin of the intestinal epithelium). Indeed, there may be different niches that may have been underappreciated in the work as it stands (niche peculiarities may depend on geo-localization, without considering the other, more posterior, perivascular niche).

Point 2 (that also relates to essential point 3 of the Evaluation Summary): The point on the APEX2 signal to unequivocally identify the HSPCs under study has not been addressed in the revision. This is important in particular for the data that illustrate the contact between dbh cells and putative HSPCs which is scarce (1 couple of cells in new Figure 7 (previous Figure 8, kept unchanged)); as it stands, this result is still preliminary (the HSPC is not identified unequivocally. The authors argue in their response that they did not manage to optimize the protocol so as to retain the GFP and mCherry signals but can't they take the peroxidase signal as reference (together with the gfp signal from dbh cells))?

[editor note: please reword the specific section to discuss other options and limitations]

Remark on essential Point 7: the answer to that point is well taken and understandable. In addition, since the strategy is to use mosaicism (hence reduced number of labelled HSPCs), this decreases the chance to capture the cells that immobilize in the perivascular, more distant niche (the HSPCS proximal to glomeruli appear to be more clusterized, increasing the chance to capture some of them). The authors have added in the discussion (lines 400-401 and not 392-394) that they are working on a labelling approach that would make it possible. I believe it is fair considering this point beyond the scope of this study.

---

## [Author Response]

Essential Revisions:The main points of agreement are the following:1) Multiple different lines of fish are used to highlight HSPCs, but since their signals do not overlap, it is hard to come to a definitive conclusion of what cells are analysed each time. IT seems that Runx1:GFP would be the best line to use, but it is actually the least used. CD41:GFP is used in some analyses too and especially in the validation of the APEX transgenic line (though not shown).

We acknowledge we have used multiple different zebrafish HSPC reporter lines throughout this study. They are all well-established HSPC reporters that have varying degrees of overlap with each other. Different lines were selected for each experiment based on their color (GFP or mCherry) and compatibility with other reagents, and if they marked a large or small proportion of the HSPC pool. In fact, we are greatly encouraged that the different HSPC lines have all produced consistent results that confirm the main conclusions from our study.

Previously, we quantified and reported the overlap between these lines in the caudal hematopoietic tissue (CHT) at 72 hours post fertilization (hpf) (Owen J Tamplin et al., 2015). We have copied that information here: “Runx:GFP overlaps 92±11% with Runx:mCherry. Runx:mCherry overlaps 13±6% with Runx:GFP. (CHT of n=11 embryos scored)” and “cd41:GFP overlaps 60±12% with Runx:mCherry. Runx:mCherry overlaps 44±8% with cd41:GFP. (CHT of n=12 embryos scored).”

For mosaic HSPC expression we used the *draculin* (*drl*) promoter because it marks HSPCs and drives expression at a higher level than other available promoters (e.g., Runx+23 or cd41). Others have validated overlap between Runx:mCherry and drl:GFP in HSPCs (Henninger et al., 2017; Mosimann et al., 2015), and showed that virtually 100% of Runx:mCherry positive cells are also drl:GFP positive from embryo to adult. One caveat of this F0 mosaic transgenic approach is that there is inherent variability between embryos. This is what we wanted because it allowed us to select embryos that had positive cells lodged in the kidney that were also sparse for single cell correlation between imaging modalities. Therefore, we selected embryos that had similar cell numbers to those in cd41:GFP embryos (i.e., 1-2 in EC pockets and 1-5 in clusters; new Figure 3 —figure supplement 1B).

Overall, we observed excellent overlap between Runx:mCherry, Runx:GFP, cd41:GFP, and draculin promoter expression in the HSPC compartment that has also been validated in other studies. Together, these lines have demonstrated HSPCs in the larval kidney marrow niche of 5 dpf zebrafish are consistently found in three locations: (1) vessel lumen; (2) clusters adjacent to the glomerulus; (3) and lodged in perivascular sites (new Table 1). We have quantified all of the data from our light sheet experiments and consistently found there are only 1-2 HSPCs in pockets, regardless of which line is used (new Figure 3 —figure supplement 1B).

Regarding the suggestion that we use the Runx:GFP line for additional experiments, over the last 2 years our Runx:GFP line has undergone transgene silencing that is a common problem with tol2 transposase-generated zebrafish lines after multiple generations. We have been working to recover the line from frozen stocks, and regenerate the line using I-SceI meganuclease transgenesis to avoid this problem in the future, but we do not yet have a line available to perform new experiments.

2) Only two examples of HSCPs surrounded by the 7 cells niche are presented. Is this the case for all HSPCs, or what sort of variation is observed? Variation would be expected given the reporters used are marking HSPCs rather than primitive HSCs, and therefore a heterogeneous population. In general, more data should be presented and statistically analysed to support the conclusions presented.

In this revised manuscript we present more data to address the variation of HSPC lodgement in the niche of larval zebrafish at 5 dpf. We have generated one new serial section EM dataset (#4 cd41:GFP).

First, to provide context for our system, HSPC lodgement in the nascent kidney marrow of the larval zebrafish at 5 dpf is a very rare event. The predicted number of HSPCs produced from the dorsal aorta of the zebrafish embryo is considerably less (~20-fold) than mouse models. Instead of hundreds of cells produced from the dorsal aorta of the mouse embryo (Ganuza et al., 2017), in zebrafish it is less than 30 (Henninger et al., 2017). This makes tracking and quantification of many events very difficult, particularly at the early stages of larval kidney niche colonization when we are tracking the arrival of the very first cells. Our Runx:mCherry line has the broadest coverage of the HSPC pool and it shows there are only 50-100 positive cells in the entire kidney marrow niche between 4-5 dpf.

Another considerable challenge with this project is acquisition of serial section electron microscopy dataset (SBEM). From initial light imaging, to embedding, microCT scans, serial section scanning, post-acquisition data alignment and analysis, takes many months of work. This explains why some high impact studies using serial section EM only present a single dataset (Hildebrand et al., 2017). In the first submission of our study, we presented three serial section EM datasets: #1 drl:APEX2-mCherry; #2 drl:APEX2-mCherry; #3 dbh:GFP. For this resubmission we have produced a new serial section EM dataset: #4 cd41:GFP (new Figure 6). This new dataset using the cd41:GFP HSPC-specific reporter line has allowed us to analyze many lodged HSPCs in a single SBEM dataset. This has allowed us to make conclusions about common features of HSPC lodgement and confirm consistent observations with our previous datasets. Although each of the four SBEM datasets was generated to address slightly different questions, we did observe and quantify common features of HSPC lodgement in the niche of each sample. We have described niche interactions for a total of n=22 putative HSPCs collected from all SBEM datasets and summarized these results in new Table 1.

3) APEX signal is not always shown, but should be used to unequivocally identify the cell presented.

We acknowledge that the APEX2 staining approach was not always used to generate our serial section EM datasets. We did not clearly explain in the first version of the manuscript that we developed two distinct correlative light and electron microscopy (CLEM) workflows. Workflow #1 was dependent on having sparse but high levels of APEX2 expression in the HSPC compartment. For that purpose, we used the draculin promoter to express mCherry (for light microscopy) and APEX2 (for EM) from the same construct. This was injected transiently in F0 embryos because we wanted sparse mosaic labeling. APEX2 is a genetically encoded peroxidase that we tagged to nuclei and mitochondria (Lam et al., 2015), allowing us to match position across imaging modalities. The light and EM 3D datasets were merged and aligned using Imaris software.

Workflow #2 was a significant technical advance because it could be used with any GFP-expressing transgenic line. The embryos were treated with DRAQ5 nuclear stain before confocal imaging. Nuclear labeling provided the landmarks that were needed in both 3D light microscopy and EM datasets for software correlation and alignment. This second workflow was more rapid and robust in its alignments, and was used to confirm the location of cd41:GFP+ cells (new Figure 6) and dbh:GFP+ cells (Figure 7) in the niche. What we found so striking about the approach developed in Workflow #2 was the precision of alignments at the single cell level (new Figure 6 —figure supplements 1 and 2), and the ability to (as stated by the reviewer) unequivocally identify any cell between light microscopy and EM modalities.

Our results were consistent using the two different workflows and identified HSPCs in three locations (as mentioned above): (1) vessel lumen; (2) clusters adjacent to the glomerulus; (3) and lodged in perivascular sites (new Table 1).

4) In figure 2 ZO-1 does not look particularly enriched around Cherry positive cells, and the relative conclusions should be rephrased.

We have edited the text in Lines 135-138 to rephrase it as shown below. Since ZO-1 staining is present on all ECs, including those ECs that surround the HSPC, our observation suggests that it could be a potential candidate for mediating connections between ECs and HSPCs.

“We observed expression of ZO-1 broadly on ECs, as well as localization between Runx:mCherry HSPCs and surrounding niche cells (Figure 2A,B). These data suggest tight junctions form at the contact points between HSPCs and the niche.”

5) Scale bars are not always clear, and a number of figure panels are hard to read (see specific comments). The same applies to multiple videos.

We have made the requested changes to figure panels and the videos.

6) One point to be better discussed is how the Authors reconcile the similar organisation of CHT and KM niches, when functionally they are very different, ie they support rapidly expanding and more quiescent HSPCs, respectively.

We have addressed this excellent point in the Discussion section with the following paragraph:

“A major outstanding question is if there is functional heterogeneity within different niche sites of the CHT and KM. For example, the 1:1 attachment of an HSPC to an MSC that we observe for 50-60% of HSPCs could regulate quiescence versus expansion. There may also be subtypes of ECs within the vasculature of the CHT and KM that provide different regulatory functions, as is observed in the adult mouse bone marrow (J. Zhang et al., 2021). The CHT of the zebrafish differs from the fetal liver because there is not the massive exponential expansion of HSPCs required to sustain the mammalian embryo in utero (Kumaravelu et al., 2002; Owen J Tamplin et al., 2015). Unlike mammalian embryos, zebrafish embryos develop externally and can survive for many days with no definitive hematopoiesis (Sood et al., 2010; Y. Zhang, Jin, Li, Qin, and Wen, 2011). We do not know if the HSPCs that transit through the CHT to the KM have equal potential to become long term quiescent HSCs, or if there is already heterogeneity. Some HSPCs may be fated to produce the blood lineages required during development, while others may retain their quiescence as they transit through the CHT to the KM. Although we don’t have the genetic tools yet to functionally distinguish between different sites of HSPC lodgement within the larval KM niche, we are developing a labeling approach that would make this possible.”

7) Is it the case that there are two anatomically and functionally distinct niches in the KM, the glomerular and perivascular one? The functional difference needs to be proven. One possibility would be to identify differential influence of dopamine on these two niches.

This is a very interesting point however the genetic tools are not yet available to address this point. We do not have a single cell labeling strategy for HSPCs in the 5 dpf larval kidney that would allow us to mark and track cells in different locations. We have added the following lines to the Discussion section (392-394):

“Although we don’t have the genetic tools yet to functionally distinguish between different sites of HSPC lodgement within the larval KM niche, we are developing a labeling approach that would make this possible.”

Reviewer #1 (Recommendations for the authors):ED1A. Could some contrast indicating the glomerulous and tubules be shown?

A new image has been added as Figure 1 —figure supplement 1A with a single z-section through the sample with better contrast for glomerulus and the tubules.

ED1I. Please show average and error bars rather than the summarised pie chart.

A plot showing average and error bars has been added and is the new Figure 1 —figure supplement 1J.

Video 1 is very hard to follow as cells and structures move significantly from one frame to the next.

We acknowledge there is lots of cell movement in this video because of circulating HSPCs, however the HSPCs in the mediolateral clusters are mostly static. We added this note to the Video 1 figure legend:

“We observed HSPCs that were rapidly moving in circulation, as well as those that were mostly static in the mediolateral clusters.”

Video 2. Perhaps the more novel finding is that most of the interactions are relatively short lived. This should be at least noted and commented on.

We added this note to the Video 2 legend:

“Occasionally, HSPCs slow and attached to vessels in the perivascular region of the larval KM niche.”

Figure 2C. Why CD41gfp and not Runx1:GFP? DsRed resolution is not sufficient to rule out that each HSPC is in touch with just one MSC.

A different image has been used in Figure 2C to show cxcl12:DsRed2+ stromal cell interaction with cd41:GFP+ HSPCs.

Line 106. Most other HSPCs were within 1 cell diameter: 4 vs 8 are further.

The text in Lines 140-142 has been edited to make this clearer:

“We measured the distance between HSPCs and MSCs and observed 57% (n=16/28) of HSPCs were in direct contact with an MSC, 29% were <5 µm (n=8/28), and 14% (n=4/28) were <10 µm away (Figure 2D).”

The characterization of the new transgenic fish line needs to be more precise and better shown. How can there be 1 or 2 HSPC/EC pockets, but many more HSPC clusters? IS there a functional difference between the single cells in EC pockets and the cells in clusters?

Regarding the F0 transient drl:APEX2-mCherry transgenics, we discussed this above (Essential Revision #1) and have copied our response here:

For mosaic HSPC expression we used the *draculin* (*drl*) promoter because it marks HSPCs and drives expression at a higher level than other available promoters (e.g., Runx+23 or cd41). Others have validated overlap between Runx:mCherry and drl:GFP in HSPCs (Henninger et al., 2017; Mosimann et al., 2015), and showed that virtually 100% of Runx:mCherry positive cells are also drl:GFP positive from embryo to adult. One caveat of this F0 mosaic transgenic approach is that there is inherent variability between embryos. This is what we wanted because it allowed us to select embryos that had positive cells lodged in the kidney that were also sparse for single cell correlation between imaging modalities. Therefore, we selected embryos that had similar cell numbers to those in cd41:GFP embryos (i.e., 1-2 in EC pockets and 1-5 in clusters; new Figure 3 —figure supplement 1B).

It is explained that the APEX-cherry cells are identified as also GFP+ in CD41+ fish (why not Runx-GFP?), but there is no CD41-GFP in Figure 3. Are the APEX-cherry cells truly HSPCs?

To verify that drl:APEX2-mCherry+ cells are HSPCs, we injected drl:APEX2-mCherry construct into cd41:GFP transgenics and we observed 15% overlap between mCherry+/GFP+ cells (Line 191).

Figure 5. How many HSPCs are observed in this kind of niche in how many fish? The reporters used are not very strict for primitive HSPCs and variability would be expected and should be shown. At the moment it cannot be claimed that 'HSPCs lodge in a highly ordered multicellular niche'.

We have described all the HSPC-niche interactions we have observed for n=22 putative HSPCs from 4 independent SBEM datasets (new Table 1). We have also reworded our conclusions as follows to describe a “multicellular niche”:

Line 264: Finally, given this multicellular HSPC niche structure we observed…

Figure 5 title: HSPCs lodge in a multicellular niche in the perivascular KM.

Figure 7 and 8. APEX staining is missing but CLEM is equally completed. Is APEX necessary then?

Our understanding is that the Reviewer is referring to Figures 6 and 7 in the previous version of the manuscript. These are now renamed as Figure 5 —figure supplement 2 and 3. These represent the same sample. We performed CLEM and aligned the APEX2+/mCherry+ HSPC. However, this HSPC was attached to a vessel wall but was not lodged in the posterior perivascular niche. Since HSPCs are generally rounded with a large nucleus, as shown previously (O. J. Tamplin et al., 2015). In Figure 5 —figure supplement 3, we looked at the SBEM data to identify other unlabeled putative HSPCs in the perivascular KM niche based on cellular morphology alone. The identification of two unlabeled putative HSPCs in the same anatomical location further confirmed that HSPCs lodge in the perivascular KM niche and interact with multiple niche cells.

In Essential Revision #3 above, we describe in detail our two CLEM workflows and how they are used to tackle different questions. Workflow #1 uses APEX2 and Workflow #2 uses any GFP+ transgenic line.

Reviewer #2 (Recommendations for the authors):1) The authors indicate that Runx:GFP is a better HSPC marker than Runx:mCherry. Yet the data with Runx:GFP are scarce. Runx:GFP is only used in Figure 1 and the less specific Runx:mCherry is used in the other main figures. Figure 1 also lacks quantification. How many HSPCs are surrounded by how many endothelial cells? Where are GFP+ cells located in relationship to other niche cells?

We have now described all the HSPC-niche interactions we have observed for n=22 putative HSPCs from 4 independent SBEM datasets (new Table 1). In Essential Revision #1 above, we describe in detail the different applications for the various transgenic lines used in this study.

2) The authors provide two examples of HSPCs surrounded by 7 cell niche (5 ECs, 1 MSC, 1 Glial like cell), but we don't know if this the case for every HSPC or just a subset/what proportion.

See (1) above.

3) Figure 7 importantly identifies putative HSPCs based on morphological criteria, but again only few examples are shown and confirmation that these are HSPCs using the reporter lines seems required.

See (1) above.

4) In Figure 8D, HSPCs have been identified as 'morphologically resembling an HSPC'; some of the reporters available should be used to demonstrate that this is the case.

Using our Workflow #2, we validated the presence of dbH^+^ cells within the KM niche by confocal imaging of dbh:GFP^+^ transgenic larva, followed by SBEM, and then correlation of the two datasets using the DRAQ5 approach. We agree with the reviewer that it would have been ideal to perform this correlation using runx:mCherry^+^/dbh:GFP^+^ or APEX2:mCherry^+^/dbh:GFP^+^ double transgenics. However, we were not able to optimize the protocol in a way that we could retain both the GFP and mCherry signals and perform SBEM. Therefore, we proceeded with dbh:GFP^+^ transgenic alone. Instead, we present light microscopy data and quantification showing that most HSPCs are in contact or close proximity to dbh:GFP+ cells (Figure 7B and Figure 7 —figure supplement 1A).

5) L. 101: "This suggests that ZO-1 is a potential candidate for mediating adhesion between HSPCs and the surrounding niche cells in the KM niche" and L. 244-245 of Discussion – Figure 2A does not show a particular enrichment of ZO-1 expression near mCherry+ cells.

Copied from above (Essential Revision #4):

We have edited the text in Lines 135-138 to rephrase it as shown below. Since ZO-1 staining is present on all ECs, including those ECs that surround the HSPC, our observation suggests that it could be a potential candidate for mediating connections between ECs and HSPCs.

“We observed expression of ZO-1 broadly on ECs, as well as localization between Runx:mCherry HSPCs and surrounding niche cells (Figure 2A,B). These data suggest tight junctions form at the contact points between HSPCs and the niche.”

6) L.226-7: "suggesting a role for dopamine signalling in colonization of the presumptive adult KM niche" should probably be rephrased. There could be different reasons and neurotransmitters/other molecules responsible for this effect.

We have revised this statement as follows (Line 350-352):

“We observed a significant reduction in the number of Runx:mCherry^+^ HSPCs within the niche after treatment with this dopaminergic cell neurotoxin (Figure 7C; Figure 7 —figure supplement 1C,D).”

7) Scale bars in some figures e.g 3D-G weren't always clear.

The scale bars have been modified.

Reviewer #3 (Recommendations for the authors):Please find my major comments beneath on Figures and related text:– Figure 3 and textThe information is not easily captured by the reader.In general, panels D-G need annotations (I would suggest: DA and ISVs in D, the position of DAB stained regions in E, the main organs that we see in F (I guess we see the notochord and part of the swim bladder?), G: the position of somites (s1, s2, s3, s4; with somite 2 nearby the kidney marrow as shown in the very nice suppl. Video 4)).More specific comments on specific panels:D: I doubt that the large, pink, elongated structure is indeed HSPCs (could it be the auto-fluorescence (AF) from the gut as seen in Extended Data Figure 3? I guess this is the same for Figure 4, panel A)

These are excellent suggestions and we have updated Figure 3D-G to include these details.

E: the local DAB precipitate(s) is not clearly visible (could the authors make attempt to obtain an image before and after DAB staining, on the same larva (and show the comparison of the 2 of them?)).

We do not have an example of a larva before DAB staining; however, we have selected an example at higher magnification with the region of the anterior KM niche labeled with a box (new Figure 3E).

For the accompanying Extended Data Figure 5, the legend of panel D says we see a reconstitution of HSPCs and multicellular niche (line 851), where are they? could the authors delineate the corresponding regions?

The data discussed previously was referring to new Figures 5 and 6. To address this comment, we have edited the figure legend for Figure 3 —figure supplement 2 as follows:

(D) Representative SBEM volume (dimensions: 31x22x30 µm).

For the Supplementary Video 5, which is truly impressive, and beautiful in resolution, with plenty of information, what a pity not to have annotations to get a maximum benefit from it; the frustration is immense! I propose that authors decompose it for the images that are the most in relation to HSPC clusters and the relevant surrounding organs (including vascular structures) and build an additional piece of Extended Data.

We have added a new Figure 5 —figure supplement 1 to show another region taken from a single frame of Video 5 (i.e., another section of the full SBEM dataset). We have also annotated the video to label landmark tissues. Upon publication the full resolution SBEM data will be made publicly available.

I realize that one of the sections is composing Figure 5A (00:15 of the pile of images), which should appear in the legend to this Figure.

The information has been added to the Figure 5 legend.

In addition, are the panels Bi-iii of Figure 5 also extracted from Supplementary Video 5 ? In addition, for Video 5, should we expect to see the two different regions in which clusters and single (or 1-2) cells are supposed to establish 2 specific (and different) niches?

Figure 5 panels Bi-iii are taken from the full resolution (9x9 nm/pixel) SBEM dataset. Video 5 was compiled from low resolution SBEM images (1/10 the original resolution) simply to provide the reader with an overview of the full dataset. New Figure 5 —figure supplement 1 shows a different section from the SBEM dataset that shows the hematopoietic cluster near the glomerulus.

Finally, it appears that red blood cells (RBCs, which are nucleated in the zebrafish) have a very dark cytoplasm 00:11 for ex ? is it DAB staining (in the cytoplasm then and not in nuclei?) or is it endogenous cytosolic peroxydase activity which is revealed here?

It is known that zebrafish erythrocytes have high endogenous peroxidase activity (Yamasaki and Nakayasu, 2003), and we believe our data shows these high cytosolic levels. To insure specificity of our genetically encoded APEX2 peroxidase, we tagged the nucleus and mitochondria, distinguishing our target cells from any other cells that may have dark contrast (Figure 5Aiii).

In the text that relates to Figure 3, I find a potential inconsistency in the intention of having sparse labelling of HSPCs (line 131), which I understand is an obvious advantage if one wants to increase the chance to visualize single cells with no other labelled contacting hematopoietic cell (and hence using F0 larvae), but to have as an outcome the same result as when using stable transgenic line (see lines 139-140) is rather unexpected. Could the authors comment on that?

This was addressed in Essential Revision #1 (copied below):

For mosaic HSPC expression we used the *draculin* (*drl*) promoter because it marks HSPCs and drives expression at a higher level than other available promoters (e.g., Runx+23 or cd41). Others have validated overlap between Runx:mCherry and drl:GFP in HSPCs (Henninger et al., 2017; Mosimann et al., 2015), and showed that virtually 100% of Runx:mCherry positive cells are also drl:GFP positive from embryo to adult. One caveat of this F0 mosaic transgenic approach is that there is inherent variability between embryos. This is what we wanted because it allowed us to select embryos that had positive cells lodged in the kidney that were also sparse for single cell correlation between imaging modalities. Therefore, we selected embryos that had similar cell numbers to those in cd41:gfp embryos (i.e., 1-2 in EC pockets and 1-5 in clusters; new Figure 3 —figure supplement 1B).

Also, do the 1-2 mCherry+ cells/EC pockets and 1-5 mCherry+/clusters were observed in each of the 11 larvae that were analyzed (owing to expected mosaicism, this may not be the case)?

As described above, there was inherent variability between embryos, but we intentionally selected F0 drl:APEX2-mCherry embryos that had HSPC labeling that was similar to cd41:GFP stable lines (see new Figure 3 —figure supplement 1B).

– Figure 4 and textFigure 4 is quite informative for comprehending the strategy followed technically but, somehow, is not illustrating the text lines 173-175 that makes the statement '… dark nuclear and mitochondrial staining … confirming that we have identified the same APEX2+/mCherry+ HSPC …'; thus, the authors should provide a magnification of that HSPC (from the EM image in panel Dii, with a clear nuclear and mitochondrial dark APEX2-derived staining).

We revised the statement to make it clear that Figure 4D only the dark nuclei is visible (Lines 224-227):

“Furthermore, the APEX2^+^ cell had dark nuclear staining with much higher contrast than any of the surrounding cells, confirming we had identified the same APEX2^+^;mCherry^+^ HSPC across multiple imaging modalities (Figure 4D).”

The data requested is presented in Figure 5 Aii and Aiii.

– Figure 5 and textI find an inconsistency saying that the posterior perivascular niche encloses a single HSPC as stated line 187 (or may be 2 maximum ? as stated lines 137 and 140 if I understand correctly, which is also considered as a 'single rare' cell in Figure 6 ) (with the ratio 1xHSPC/5xEC/1xMSC) when the figure (see also the legend of Biii lines 709-710) shows two additional – non labelled – HSPCs (HSPCs 2 and 3); can the authors comment on that ?

Thank you for the comment. We understand clarification is necessary. We believe this relates to Essential Revision #2 and have copied the response below:

We have described niche interactions for a total of n=22 putative HSPCs collected from all SBEM datasets and summarized these results in new Table 1.

Supplementary Video 6 is impressive, but again (in particular for the planes 00:00 to 00:33), some annotations should be added on most relevant tissues/organs/cell types.

We added annotations to Video 6 at 00:05 (the frame that corresponds to Figure 5) to outline relevant tissues.

In addition, and on conceptual grounds, it is proposed that the CHT niche(s) is (are), in their vast majority, the site(s) of progenitors expansion, meaning most probably a very minor proportion would home cells bearing/maintaining full, long-term HSC potential (as opposed to the kidney marrow in which future adult HSCs are homing, thus with an expected higher proportion of HSC-specific niches). Based on this, how would the authors conceive that such conserved structures would accommodate such differentially fated cells (see the statement lines 187-188 'demonstrating this cellular structure is conserved between developmental and presumptive adult niche')?

This is similar to Essential Revision #6 and we have copied our response below:

We have addressed this excellent point in the Discussion section with the following paragraph:

“A major outstanding question is if there is functional heterogeneity within different niche sites of the CHT and KM. For example, the 1:1 attachment of an HSPC to an MSC that we observe for 50-60% of HSPCs could regulate quiescence versus expansion. There may also be subtypes of ECs within the vasculature of the CHT and KM that provide different regulatory functions, as is observed in the adult mouse bone marrow (J. Zhang et al., 2021). The CHT of the zebrafish differs from the fetal liver because there is not the massive exponential expansion of HSPCs required to sustain the mammalian embryo in utero (Kumaravelu et al., 2002; Owen J Tamplin et al., 2015). Unlike mammalian embryos, zebrafish embryos develop externally and can survive for many days with no definitive hematopoiesis (Sood et al., 2010; Y. Zhang et al., 2011). We do not know if the HSPCs that transit through the CHT to the KM have equal potential to become long term quiescent HSCs, or if there is already heterogeneity. Some HSPCs may be fated to produce the blood lineages required during development, while others may retain their quiescence as they transit through the CHT to the KM. Although we don’t have the genetic tools yet to functionally distinguish between different sites of HSPC lodgement within the larval KM niche, we are developing a labeling approach that would make this possible.”

– Figure 8 and textThe Figure with the light sheet information on close proximity between HSPCs and dbh cells is nice and the quantitative analysis panel (C) convincing.However, the data should be substantiated using a double Tg line (draculin:2APEX2-mCherry) X (dbh:GFP) to confirm the identity of the HSPC(s) contacting the dbh cell(s) and not only a resemblance as state line 231 (hence labelled by the APEX2 nuclear + mitochondrial activities). This would very much substantiate the work and strengthen the manuscript on this important aspect of HSPC/niche establishment/regulation (and the authors have all the tools/approaches at hand). This is also required since the authors clearly state at the end of the Discussion (line 262) '… additional rounds of CLEM to confirm that dbh:GFP+ cells were adjacent to HSPCs …'; clearly, identifying unambiguously HSPCs is mandatory.

This is similar to Reviewer #2 comment #4 above and we have copied our response below:

Using our Workflow #2, we validated the presence of dbH^+^ cells within the KM niche by confocal imaging of dbh:GFP^+^ transgenic larva, followed by SBEM, and then correlation of the two datasets using the DRAQ5 approach. We agree with the reviewer that it would have been ideal to perform this correlation using Runx:mCherry^+^/dbh:GFP^+^ or APEX2:mCherry^+^/dbh:GFP^+^ double transgenics. However, we were not able to optimize the protocol in a way that we could retain both the GFP and mCherry signals and perform SBEM. Therefore, we proceeded with dbh:GFP^+^ transgenic alone. Instead, we present light microscopy data and quantification showing that most HSPCs are in contact or close proximity to dbh:GFP+ cells (Figure 7B and Figure 7 —figure supplement 1A).

In the accompanying Extended Data Figure 9, panel (A) line 892, what do (n=10) refers to? 10 sections of 1 experiment? 1 section, each from 10 independent experiments? What cells does the blue colour in panel Bii underline?

n=10 indicates the number of larvae in which cells were counted. Blue cells are DRAQ5 labeled nuclei. This is clarified in the revised Figure 7 —figure supplement 1 legend (relevant section copied below):

“Numbers above the columns indicate the cell numbers counted in each group (combined data from n=10 larvae). (B) 3D CLEM alignment of confocal dataset with DRAQ5 labeled nuclei (blue in (i), (ii), and (iv)) and SBEM datasets identifies a dbh:GFP^+^ cell in the KM niche. (i) Alignment between confocal and SBEM datasets in the XY plane. (ii) White arrowhead points to the single dbh:GFP^+^ cell in the aligned confocal and SBEM datasets, and in (iii) the green outlined dbh:GFP^+^ cell in SBEM data alone. (iv) dbh:GFP^+^ cell in confocal data alone.”

In the text, the authors say that they wish to address a role of dopamine signalling in the KM niche colonization (hence they treat with 6-OHDA from 4 to 5dpf, during niche colonization); would a shorter treatment (ex: few hours) also affect HSPCs in the niche, which would indicate a function of dopamine in the niche per se (ex: survival, maintenance) rather than colonization?

We performed drug treatments for shorter durations (2 hours and 4 hours) starting at 120 hpf (5 dpf) and we did not see an effect on HSPCs in the niche (data not shown). We are pursuing the role of dopamine signaling in the niche in more detail in a future study that we believe is beyond the scope of this current manuscript.

Finally, the authors propose, if I understand correctly, that there may be 2 different types of functional niches (1) proximal/adjacent to the glomerulus, (2) more distant and referred to as posterior perivascular niche; do they have any information on a possible differential influence of dopamine and dbh-cells on these 2 niche types (or to their colonization)? Answering to that question would as well strengthen the functional impact of the work.

This is similar to Essential Revision #7 and we have copied the response below:

This is a very interesting point however the genetic tools are not yet available to address this point. We have added the following lines to the Discussion section (393-395):

“Although we don’t have the genetic tools yet to functionally distinguish between different sites of HSPC lodgement within the larval KM niche, we are developing a labeling approach that would make this possible.”

Comments on the Discussion:The authors should, in my opinion, discuss the possible functional discrepancies between the 2 geographic positions of the niches that are visualized here, i.e the one adjacent to the glomerulus and the one that is perivascular (more distant from the glomerulus); they should do so also in the context of the current work on mammalian bone marrow niches. This is an important issue in the field because, ultimately, one wants to comprehend what are the functional differences between the niches that allow the maintenance of long-term HSCs and the ones that are more devoted to support specific lineage differentiation. Do the authors have a vision on that issue?

This is similar to Essential Revision #6 and again we have copied our response below:

We have addressed this excellent point in the Discussion section with the following paragraph:

“A major outstanding question is if there is functional heterogeneity within different niche sites of the CHT and KM. For example, the 1:1 attachment of an HSPC to an MSC that we observe for 50-60% of HSPCs could regulate quiescence versus expansion. There may also be subtypes of ECs within the vasculature of the CHT and KM that provide different regulatory functions, as is observed in the adult mouse bone marrow (J. Zhang et al., 2021). The CHT of the zebrafish differs from the fetal liver because there is not the massive exponential expansion of HSPCs required to sustain the mammalian embryo in utero (Kumaravelu et al., 2002; Owen J Tamplin et al., 2015). Unlike mammalian embryos, zebrafish embryos develop externally and can survive for many days with no definitive hematopoiesis (Sood et al., 2010; Y. Zhang et al., 2011). We do not know if the HSPCs that transit through the CHT to the KM have equal potential to become long term quiescent HSCs, or if there is already heterogeneity. Some HSPCs may be fated to produce the blood lineages required during development, while others may retain their quiescence as they transit through the CHT to the KM. Although we don’t have the genetic tools yet to functionally distinguish between different sites of HSPC lodgement within the larval KM niche, we are developing a labeling approach that would make this possible.”

[Editors’ note: further revisions were suggested prior to acceptance, as described below.]

Reviewer #2 (Recommendations for the authors):The manuscript describing this technically groundbreaking and very impressive approach to study the niche is largely improved and provides a more balanced interpretation of the results, considering the inherent limitation of the approach including datasets available.I understand that the approach is technically very challenging and time-consuming, and would not allow to obtain larger datasets to perform quantitative analyses and obtain more definitive conclusions. Therefore, these should be rephrased to emphasise the qualitative (rather than quantitative) nature of the analysis and acknowledge some limitations (such as the lack of markers for putative HSCs in some cases). The contact with glial cells is emphasised but it should be noted that only 3 out of 22 HSPCs were found in contact with glial cells. Besides the numerical difference of HSPCs found in contact with different niche cells, whether HSPCs contacting these different niche cells might be functionally distinct could be incorporated into the discussion. The conclusions should acknowledge the qualitative nature of the study and emphasise the strengths and limitations of this novel approach to study the niche in the future, since it could potentially elicit significant interest in the field to use it in different experimental settings.

We thank the reviewer for their comments. We have updated the text to emphasize the qualitive nature of the approach and acknowledge some limitations along with the strengths. We will discuss that only 3 out of 22 HSPCs were found in contact with glial cells. Also, we have discussed that HSPCs could be functionally distinct based on their contact with different niche cells.

Reviewer #3 (Recommendations for the authors):Comments to the AuthorsMany thanks to the authors to have provided a significant revision of their work; obviously, this correlative multimodal microscopy work is a tedious, difficult, time-consuming but essential task.While many of my points have been taken into account, there still remains 2 critical aspects that I believe should be addressed and that are partly emphasized by the new dataset (dataset 4 using the CD41:gfp line, workflow 2).Point 1 (that also relates to essential point 2 of the Evaluation Summary): The authors have provided a new dataset (see new Figure 6 and the 2 Supplemental Figures+ Table 1) that significantly increases the number of HSPCs that are analysed, including the identification of potential niche contacting cells (Table 1). However, the dataset raises the critical point of heterogeneity of the niche environment because the cluster of HSPCs that is analysed appears disconnected to any glial-like cells (GL, see outcome = 0 in Table 1 for the 17 HSPCs). Hence, this new dataset strengthens the idea of the heterogeneity which is not quantitatively appreciated in the work.

We thank the reviewer for their comments. This is a similar point as stated by Reviewer #2 and we have copied the answer here:

We have updated the text to emphasize the qualitive nature of the approach and acknowledge some limitations along with the strengths. We will discuss that only 3 out of 22 HSPCs were found in contact with glial cells. Also, we have discussed that HSPCs could be functionally distinct based on their contact with different niche cells.

Also, on qualitative aspects, it appears that the HSPC cluster is more distal to the glomeruli than the HSPCs illustrated in Figure 5 and Figure 5 – figure supplement 1 and 3 for example (which appear from the images to be more proximal to the glomerulus while for new Figure 6 the cluster appears to be connecting the pronephric tubule and the margin of the intestinal epithelium). Indeed, there may be different niches that may have been underappreciated in the work as it stands (niche peculiarities may depend on geo-localization, without considering the other, more posterior, perivascular niche).

We thank the reviewer for their thoughtful observations. Regarding the observation that the HSPC cluster in new Figure 6 is less proximal to the glomerulus compared to the HSPC cluster of Figure 5, we believe it is simply the slices we have chosen to present for these figures. Depending on the oblique angle chosen for visualization through the 3D serial section blockface electron microscopy (SBEM) dataset, the glomerulus and/or pronephric tubules may or may not be visible. In Figure 4 —figure supplement 2, it is easier to see that the HSPC clusters are always located between the glomerulus and pronephric tubules. Visualization of the data aside, it is possible that depending on where in the cluster a single HSPC resides, it may be closer to a particular cell type or signal that differentially regulates its function. We will make all datasets publicly available so readers can explore them further and perform new analyses.

Point 2 (that also relates to essential point 3 of the Evaluation Summary): The point on the APEX2 signal to unequivocally identify the HSPCs under study has not been addressed in the revision. This is important in particular for the data that illustrate the contact between dbh cells and putative HSPCs which is scarce (1 couple of cells in new Figure 7 (previous Figure 8, kept unchanged)); as it stands, this result is still preliminary (the HSPC is not identified unequivocally). The authors argue in their response that they did not manage to optimize the protocol so as to retain the GFP and mCherry signals but can't they take the peroxidase signal as reference (together with the gfp signal from dbh cells)?[editor note: please reword the specific section to discuss other options and limitations]

We acknowledge that the putative HSPC in Figure 7 is not identified unequivocally based on its own HSPC-specific marker (e.g., cd41, Runx1+23, or draculin). We have been careful to describe unlabeled HSPCs as “putative”. We have emphasized that the putative HSPC in question is described based on morphology alone. For this dataset we focused on localization of the dbh:gfp+ cells to the kidney niche. Unfortunately, we cannot use the peroxidase signal generated in Workflow #1 and #2 at the same time because one relies on APEX2 and the other on DRAQ5 to create high contrast in the EM datasets and would generate overlapping signals.

Remark on essential Point 7: the answer to that point is well taken and understandable. In addition, since the strategy is to use mosaicism (hence reduced number of labelled HSPCs), this decreases the chance to capture the cells that immobilize in the perivascular, more distant niche (the HSPCS proximal to glomeruli appear to be more clusterized, increasing the chance to capture some of them). The authors have added in the discussion (lines 400-401 and not 392-394) that they are working on a labelling approach that would make it possible. I believe it is fair considering this point beyond the scope of this study.

We appreciate the reviewer acknowledging the technical challenges associated with aspects of the study.

For the reviewers’ information, we made one additional change that was not raised in the reviews. After careful consideration, we have revised the terminology of the “glial-like” cell to “ganglion-like cell” as it is a more accurate prediction of the dbh:gfp+ cell identity based on the available literature (1-4). The tyrosine hydroxylase/dopamine β-hydroxylase positive cells in that region of the embryo have been described as the “cervical sympathetic ganglion complex” (3), “superior cervical ganglion” (5), or “sympathetic cervical complex” made up of “neural crest-derived peripheral neurons, cranial ganglion neurons and glia” (1).

References

1. Stewart RA, Arduini BL, Berghmans S, George RE, Kanki JP, Henion PD, Look AT. Zebrafish foxd3 is selectively required for neural crest specification, migration and survival. Developmental Biology. 2006;292(1):174-88. doi: 10.1016/j.ydbio.2005.12.035. PubMed PMID: 16499899.

2. Stewart RA, Look AT, Kanki JP, Henion PD. Development of the peripheral sympathetic nervous system in zebrafish. Methods in cell biology. 2004;76:237-60. PubMed PMID: 15602879.

3. An M, Luo R, Henion PD. Differentiation and maturation of zebrafish dorsal root and sympathetic ganglion neurons. Annals of the New York Academy of Sciences. 2002;446(3):267-75. doi: 10.1002/cne.10214.

4. Guo S, Brush J, Teraoka H, Goddard A, Wilson SW, Mullins MC, Rosenthal A. Development of noradrenergic neurons in the zebrafish hindbrain requires BMP, FGF8, and the homeodomain protein soulless/Phox2a. Neuron. 1999;24(3):555-66. Epub 1999/12/14. doi: 10.1016/s0896-6273(00)81112-5. PubMed PMID: 10595509.

5. Zhu S, Lee J-S, Guo F, Shin J, Perez-Atayde AR, Kutok JL, Rodig SJ, Neuberg DS, Helman D, Feng H, Stewart RA, Wang W, George RE, Kanki JP, Look AT. Activated ALK collaborates with MYCN in neuroblastoma pathogenesis. Cancer cell. 2012;21(3):362-73. doi: 10.1016/j.ccr.2012.02.010. PubMed PMID: 22439933; PMCID: PMC3315700.